# Stratospheric ozone intrusion events and their impacts on tropospheric ozone in the Southern Hemisphere

Jesse W. Greenslade[1], Simon P. Alexander[2,3], Robyn Schofield[4,5], Jenny A. Fisher[1,6], and Andrew K. Klekociuk[2,3]

[1]Centre for Atmospheric Chemistry, School of Chemistry, University of Wollongong, Australia
[2]Australian Antarctic Division, Hobart, Australia
[3]Antarctic Climate and Ecosystems Co-operative Research Centre, Hobart, Australia
[4]School of Earth Sciences, University of Melbourne, Australia
[5]ARC Centre of Excellence for Climate System Science, University of New South Wales, Australia
[6]School of Earth & Environmental Sciences, University of Wollongong, Australia

*Correspondence to:* Jesse Greenslade (jwg366@uowmail.edu.au)

**Abstract.** Stratosphere-to-troposphere transport (STT) provides an important natural source of ozone to the upper troposphere, but the characteristics of STT events in the southern hemisphere extra-tropics and their contribution to the regional tropospheric ozone budget remain poorly constrained. Here, we develop a quantitative method to identify STT events from ozonesonde profiles. Using this method we estimate the seasonality of STT events and quantify the ozone transported across the tropopause over Davis ($69°$ S, 2006-2013), Macquarie Island ($54°$ S, 2004-2013), and Melbourne ($38°$ S, 2004-2013). STT seasonality is determined by two distinct methods: a Fourier bandpass filter of the vertical ozone profile, and an analysis of the Brunt-Väisälä frequency. Using a bandpass filter on 7–9 years of ozone profiles from each site provides clear detection of STT events, with maximum occurrences during summer and minimum during winter above all three sites. The majority of tropospheric ozone enhancements from STT events occur within 2.5 km and 3 km of the tropopause at Davis, and Macquarie Island. Events are more spread out at Melbourne, occurring frequently up to 6 km from the tropopause. The mean fraction of total tropospheric ozone attributed to STT during STT events is $\sim 1.0$–3.5% at each site; however, during individual events over 10% of tropospheric ozone may be directly transported from the stratosphere. The cause of STTs is determined to be largely due to synoptic low pressure frontal systems, determined using coincident ERA-Interim reanalysis meteorological data. Ozone enhancements can also be caused by biomass burning plumes transported from Africa and South America, which are apparent during austral winter and spring, and are determined using satellite measurements of CO.

To provide regional context for the ozonesonde observations, we use the GEOS-Chem chemical transport model, which is too coarsely resolved to distinguish STT events but is able to accurately simulate the seasonal cycle of tropospheric ozone columns over the three southern hemisphere sites. Combining the ozonesonde-derived STT event characteristics with the simulated tropospheric ozone columns from GEOS-Chem, we estimate STT ozone flux near the three sites and see austral summer dominated yearly amounts of between 5.7 and $8.7 \times 10^{17}$ molecules cm$^{-2}$ a$^{-1}$.

# 1 Introduction

Tropospheric ozone constitutes only 10% of the total ozone column but is an important oxidant and greenhouse gas which is toxic to life, harming natural ecosystems and reducing agricultural productivity. Over the industrial period, increasing tropospheric ozone has been estimated to exert a radiative forcing (RF) of 365 mWm$^{-2}$ (Stevenson et al., 2013), equivalent to a quarter of the $CO_2$ forcing (Forster et al., 2007). While much tropospheric ozone is produced photochemically from anthropogenic and natural precursors, downward transport from the ozone-rich stratosphere provides an additional natural source of ozone that is particularly important in the upper troposphere (Jacobson and Hansson, 2000, and references therein). The contribution of this source to overall tropospheric ozone budgets remains uncertain (Škerlak et al., 2014), especially in the southern hemisphere. Models show stratospheric ozone depletion has propagated to the upper troposphere (Stevenson et al., 2013). However, work based on the Southern Hemisphere Additional OZonesonde (SHADOZ) network suggests stratospheric mixing may be increasing upper tropospheric ozone near southern Africa (Liu et al., 2015; Thompson et al., 2014). Uncertainties in the various processes which produce tropospheric ozone limit predictions of future ozone-induced radiative forcing. Here we use a multi-year record of ozonesonde observations from sites in the southern hemisphere extra-tropics, combined with a global model, to better characterise the impact of stratospheric ozone on the tropospheric ozone budget in the southern hemisphere.

Stratosphere-to-troposphere transport (STT) primarily impacts the ozone budget in the upper troposphere but can also increase regional surface ozone levels above the legal thresholds set by air quality standards (Danielsen, 1968; Lelieveld et al., 2009; Lefohn et al., 2011; Langford et al., 2012; Zhang et al., 2014; Lin et al., 2015). In the western US, for example, deep STT events during spring can add 20-40 ppbv of ozone to the ground-level ozone concentration, which can provide over half the ozone needed to exceed the standard set by the U.S. Environmental Protection Agency (Lin et al., 2012, 2015). Another important region of STT is the Middle East, where surface ozone exceeds values of 80 ppbv in summer, with 10 ppbv from STT contribution (Lelieveld et al., 2009). Estimates of the overall contribution of STT to tropospheric ozone vary widely (e.g. Galani, 2003; Stohl et al., 2003; Stevenson et al., 2006; Lefohn et al., 2011). Early work based on two photochemical models showed that 25-50% of the tropospheric ozone column can be attributed to STT events globally, with most contribution in the upper troposphere (Stohl et al., 2003). In contrast, a more recent analysis of the Atmospheric Chemistry and Climate Model Inter-comparison Project (ACCMIP) simulations by Young et al. (2013) found that STT is responsible for $540 \pm 140$ Tg yr$^{-1}$, equivalent to $\sim$11% of the tropospheric ozone column, with the remainder produced photochemically (Monks et al., 2015). This wide range in model estimates exists in part because STT is challenging to be accurately represented, and finer model resolution is necessary to simulate small scale turbulence. Observation-based process studies are therefore key in determining the relative frequency of STT events, with models then able to quantify STT impact over large regions. Ozonesondes are particularly valuable for this purpose as they provide multi-year datasets with high vertical resolution.

Lower stratospheric and upper tropospheric ozone concentrations are highly correlated (Terao et al., 2008), suggesting mixing across the tropopause mainly associated with the jet streams over the ocean. Extra-tropical STT events most commonly occur during synoptic-scale tropopause folds (Sprenger et al., 2003; Tang and Prather, 2012; Frey et al., 2015) and are char-

acterised by tongues of high potential vorticity (PV) air descending to lower altitudes. As these tongues become elongated, filaments disperse away from the tongue and mix irreversibly into the troposphere. STT can also be induced by deep overshooting convection (Frey et al., 2015), tropical cyclones (Das et al., 2016) and mid-latitude synoptic scale disturbances (e.g. Stohl et al., 2003; Mihalikova et al., 2012). STT events have been observed in tropopause folds around both the polar front jet

(Vaughan et al., 1993; Beekmann et al., 1997) and the subtropical jet (Baray et al., 2000). A big influence on the high surface ozone concentrations over the eastern Mediterranean (EM) can be attributed to stratospheric mixing and anticyclonic subsidence (Zanis et al., 2014; Akritidis et al., 2016). The EM frequently shows a summer maximum of stratospheric subsidence, which sits between 20°E and 35°E and 31°N to 39°N, while zonally most subtropical tropopause folds occur during winter (Tyrlis et al., 2014, and references therein). They are also observed near cut-off lows (Price and Vaughan, 1993; Wirth, 1995),

so both regional weather patterns and stratospheric mixing are important to understand for STT analysis.

Stratospheric ozone intrusions undergo transport and mixing, with up to half of the ozone diffusing within 12 hours following descent from the upper troposphere (Trickl et al., 2014). The long range transport of enhanced ozone can be facilitated by shear upper tropospheric winds, with remarkably little convective mixing, as shown by Trickl et al. (2014) who measure STT air masses two days and thousands of kilometres from their source. Cooper et al. (2004) also shows how STT advection can

transport stratospheric air over long distances, with a modelled STT event spreading from the northern Pacific to the East coast of the USA over a few days.

The strength (ozone enhancement above background levels), horizontal scale, vertical depth, and longevity of these intruding ozone tongues vary with wind direction and strength, topography, and season. While the frequency, seasonality, and impacts of STT events have been well described in the tropics and northern hemisphere (NH), observational estimates from the southern

hemisphere (SH) extra-tropics are noticeably absent in the literature. The role of STT in the SH remains highly uncertain due to the more limited data availability compared to the NH and the temporal sparsity of these datasets (Mze et al., 2010; Thompson et al., 2014; Liu et al., 2015).

Here, we characterise the seasonal cycle of STT events and quantify their contribution to the SH extra-tropical tropospheric ozone budget using nearly a decade of ozonesonde observations from three locations around the Southern Ocean spanning

latitudes from 38°S-69°S. Section 2 describes the observations and methods used to identify STT events and to relate STT occurrence to meteorological events. We describe how possible biomass burning smoke plume influence is detected and handled, and we introduce the GEOS-Chem model which is used for ozone flux estimation. Section 3 describes the seasonality, altitude, depth, and frequency of detected STT events, along with a comparison of our findings to other literature where possible. Section 4 analyses how well GEOS-Chem captures the tropospheric ozone seasonality and quantity near our three sites. Section 5

uses the STT detection frequencies along with GEOS-Chem monthly tropospheric ozone columns to estimate STT ozone flux near our three sites. We also compare and contrast our results against relevant literature. Finally, in Section 6 we examine in detail the uncertainties involved in our STT event detection technique and ozone flux estimations.

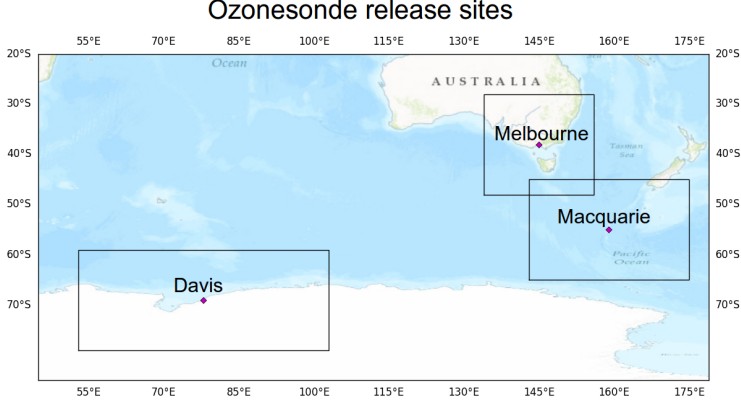

**Figure 1.** Ozonesonde release sites and the regions which will be used to examine STT effect on tropospheric ozone levels.

## 2 Data and Methods

### 2.1 Ozonesonde record in the Southern Ocean

Ozonesondes provide a high vertical resolution profile of ozone, temperature, pressure, and humidity from the surface to 35 km. In the troposphere, the ozonesondes generally make 150-300 measurements. Ozone mixing ratio is quantified with an elec-trochemical concentration cell, using standardised procedures when constructing, transporting, and releasing the ozonesondes (http://www.ndsc.ncep.noaa.gov/organize/protocols/appendix5/). Ozonesondes are estimated to provide around 2% precision in the stratosphere, which improves at lower altitudes, and ozonesondes have been shown to be accurate to within 5% when the correct procedures are followed (Smit et al., 2007).

Ozonesondes are launched approximately weekly from Melbourne (38° S, 145° E), Macquarie Island (55° S, 159° E) and Davis (69° S, 78° E), as shown in Fig. 1. Melbourne is a major city in the south east of Australia, and may be affected by anthropogenic pollution in the lower troposphere. Macquarie Island is isolated from the Australian mainland, situated in the remote Southern Ocean and unlikely to be affected by any local pollution events. Davis is on the coast of Antarctica and also unlikely to experience the effects of anthropogenic pollution.

For this study, we use the 2004-2013 data for Melbourne and Macquarie Island and the 2006-2013 data for Davis because both ozone and geopotential height (GPH) are available from the World Ozone and Ultraviolet Data Centre archived data in these periods. At Davis, ozonesondes are launched twice as frequently during the ozone hole season and preceding months (June-October) as at other times of year (Alexander et al., 2013). A summary of ozonesonde releases at each site can be seen in Table 1.

Characterisation of STT events requires a clear definition of the tropopause. Two common tropopause height definitions are the standard lapse rate tropopause (WMO, 1957) and the ozone tropopause (Bethan et al., 1996). The lapse rate tropopause is defined as the lowest altitude where the lapse rate (vertical gradient of temperature) is less than 2°C km$^{-1}$, provided the lapse

**Table 1.** Number of sonde releases at each site over the period of analysis.

| Site | Total Releases | Monthly Releases (J, F, M, ...) | Date Range |
|---|---|---|---|
| Davis | 240 | 11, 12, 13, 12, 17, 31, 29, 28, 32, 28, 15, 12 | 2006/04/13 - 2013/11/13 |
| Macquarie Island | 390 | 32, 31, 45, 28, 34, 33, 28, 35, 29, 33, 31, 31 | 2004/01/20 - 2013/01/09 |
| Melbourne | 456 | 31, 38, 40, 38, 41, 36, 38, 39, 46, 40, 38, 31 | 2004/01/08 - 2013/12/18 |

rate averaged between this altitude and 2 km above is also below $2°C\ km^{-1}$. The ozone tropopause is defined as the lowest altitude satisfying the following three conditions for the ozone mixing ratio (OMR) (Bethan et al., 1996):

1. Vertical gradient of OMR is greater than 60 ppb $km^{-1}$;

2. OMR is greater than 80 ppb; and

3. OMR exceeds 110 ppb between 500 m and 2000 m above the altitude under inspection (modified to between 500 m and 1500 m in the Antarctic, including the site at Davis).

The ozone tropopause may misdiagnose the real tropopause altitude during stratosphere-troposphere exchange; however, it is useful at polar latitudes in winter, where the lapse-rate definition may result in artificially high tropopause values (Bethan et al., 1996; Tomikawa et al., 2009; Alexander et al., 2013). We require lapse rate tropopauses to be at a minimum of 4 km altitude. Another commonly used tropopause definition (the dynamical tropopause) is determined from the potential vorticity unit (PVU; 1 PVU$= 10^{-6}\ m^2\ s^{-1}\ K\ kg^{-1}$). In the extra-tropics the isosurface where PVU$= 2$ is often used, which allows a 3D view of folds and other tropopause features in a sufficiently resolved model (Škerlak et al., 2014; Tyrlis et al., 2014). The PVU is not calculable using the ozonesonde measurements alone, so in this work the ozone tropopause is used when determining STT events or measured tropopause altitude.

Figure 2 shows the monthly median ozone tropopause altitudes at each location (solid lines). At Melbourne, the tropopause altitude displays a seasonal cycle with maximum in summer and minimum is winter. This seasonality is missing at Macquarie Island and almost reversed at Davis, which has a minimum during autumn and maximum from winter to spring. Tropopause altitude decreases with latitude from 9-14 km at Melbourne (38° S) to 7-9 km at Davis (69° S).

Figure 3 shows multi-year averaged ozone mixing ratios measured by ozonesonde over the three stations. Over Melbourne, increased ozone extending down through the troposphere is apparent from December to March and from September to November. The increased tropospheric ozone in these months is due to STT (in summer), and possible biomass burning influence (in spring), both discussed in more detail in the following sections. Over Davis and Macquarie Island, tropospheric ozone is higher between March and October, although the seasonal differences are small compared to those at Melbourne. The seasonality shown in Fig. 3 for Davis is consistent with remote free tropospheric photochemistry determined by solar radiation

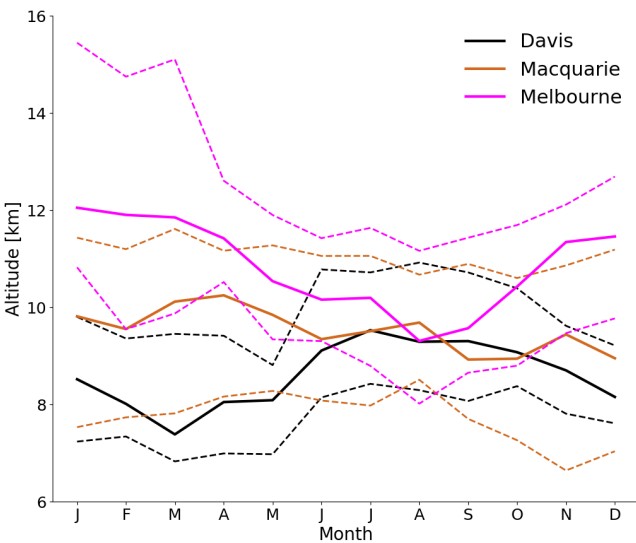

**Figure 2.** Multi-year monthly median tropopause altitude (using the ozone defined tropopause) determined from ozonesondes measurements at Davis (2006-2013), Macquarie Island (2004-2013), and Melbourne (2004-2013) (solid lines). Dashed lines show the 10th to the 90th percentile of tropopause altitude for each site.

availability and temperature, resulting in higher ozone in winter (Lelieveld and Dentener, 2000). $NO_2$ stratospheric observations have been conducted in the Southern hemisphere at Lauder, Macquarie Island and Arrival Heights (i.e. Struthers et al., 2004) which displays a winter minima in seasonality consistent with an ozone maxima. Influence from the ozone hole can be seen over Davis in October, with relatively low ozone levels extending up 5-6 km into the stratosphere.

## 2.2 Model description

To provide regional and global context to the ozonesonde observations, we use the GEOS-Chem version 10-01 global chemical transport model (Bey et al., 2001), which simulates ozone along with more than 100 other trace gases throughout the troposphere and stratosphere. Stratosphere-troposphere coupling is calculated using the stratospheric unified chemistry extension (UCX) (Eastham et al., 2014). Transport is driven by assimilated meteorological fields from the Goddard Earth Observing System (GEOS-5) maintained by the Global Modeling and Assimilation Office (GMAO) at NASA. Ozone precursor emissions are from the Model of Emissions of Gases and Aerosols from Nature (MEGAN) version 2.1 (Guenther et al., 2012) for biogenic emissions, the Emissions Database for Global Atmospheric Research (EDGAR) version 4.2 for anthropogenic emissions, and the Global Fire Emissions Database (GFED4) inventory (Giglio et al., 2013) for biomass burning emissions. Our simulation was modified from the standard v10-01 to fix an error in the treatment of ozone data from the Total Ozone Mapping Spec-

# Multi-year average ozone

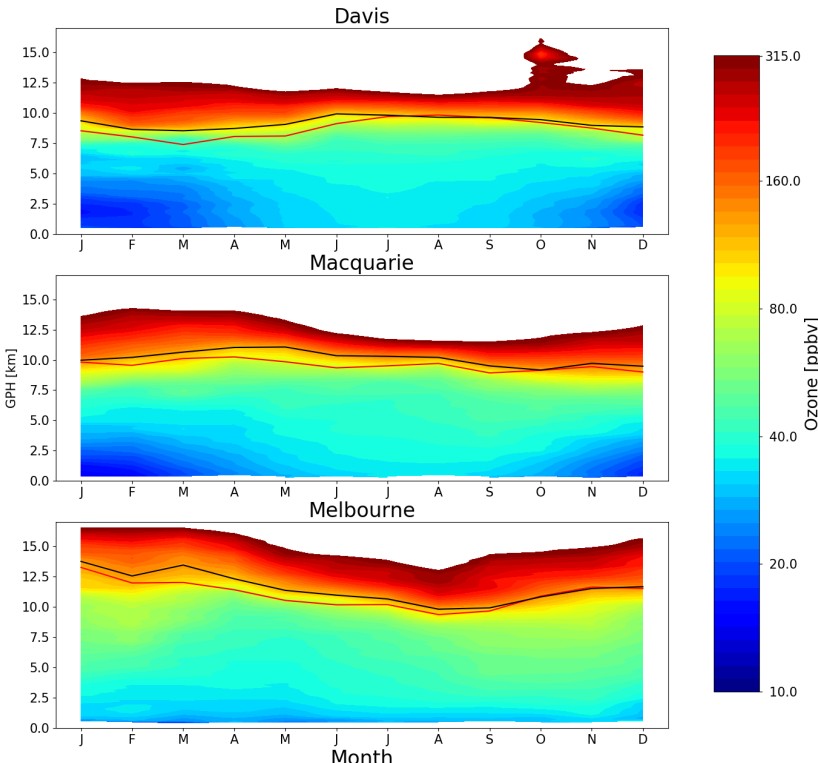

**Figure 3.** Multi-year mean seasonal cycle of ozone mixing ratio over Davis, Macquarie Island, and Melbourne as measured by ozonesondes. Measurements were interpolated to every 100 m and then binned monthly. Black and red solid lines show median ozone and lapse-rate defined tropopause altitudes (respectively), as defined in the text.

trometer (TOMS) satellite used to calculate photolysis (see http://wiki.seas.harvard.edu/geos-chem/index.php/FAST-JX_v7.0_photolysis_mechanism#Fix_for_TOMS_to_address_strange_cycle_in_OH_output.).

Our simulations span 2005-2012 (following a 1-year spin-up) with horizontal resolution of 2° latitude by 2.5° longitude and 72 vertical levels from the surface to 0.01 hPa. The vertical resolution is finer near the surface at $\sim 60$ m between levels, spreading out to $\sim 500$ m near 10 km altitude. For comparison to the ozonesonde observations, the model state was saved every 6 hours within the grid boxes containing each site. When comparing against ozonesondes, GEOS-Chem UTC+0 time samples are used for all sites. This means that the simulated ozone profiles are analysed at local times of 7AM for Davis, and 11AM for Macquarie Island and Melbourne. GEOS-Chem uses the tropopause height provided by GEOS-5 meteorological fields, which are calculated using a lapse-rate tropopause definition using the first minimum above the surface in the function $0.03 \times T(p) - \log(p)$, with p in hPa (Rienecker, 2007).

## 2.3 Characterisation of STT events and associated fluxes

We characterise STT events using the ozonesonde vertical profiles to identify tropospheric ozone enhancements above a local background (in moles per billion moles of dry air, referred to here as ppb). The process is illustrated in Figure 4 on an example ozone profile. First, the ozone vertical profiles are linearly interpolated to a regular grid with 20 m resolution from the surface to 14 km altitude. Small vertical-scale fluctuations in ozone, which are captured by the high-resolution ozonesondes, can be regarded as sinusoidal waves superimposed on the large vertical scale background tropospheric ozone. As such, the interpolated profiles are bandpass-filtered using a fast Fourier transform (Press et al., 1992) to retain these small vertical scales, between 0.5 km and 5 km (removing low and high frequency perturbations). The high frequency perturbations are removed as they may represent noise in the measurements. The perturbations with scales longer than 5 km represent the vertical gradient of ozone concentration from the surface to the stratosphere. In what follows, these filtered vertical profiles are referred to as perturbation profiles.

For an event to qualify as STT, a clear increase above the background ozone level is needed, as a bandpass filter leaves us with enhancements minus any noise or seasonal scale vertical profile effects. We next use all the perturbation profiles at each site to calculate the 95th percentile perturbation value for the site. The threshold is calculated from all the interpolated filtered values between 2 km above the surface and 1 km below the tropopause. This is our threshold for tropospheric ozone perturbations, and any profiles with perturbations exceeding this value in individual ozonesondes are classified as STT events. STT events at altitudes below 4 km are removed to avoid surface pollution, and events within 0.5 km of the tropopause are removed to avoid false positives induced by the sharp transition to stratospheric air. We note that this ozone detection methodology detailed above does not allow us to resolve STT events where the ozone flux is spread diffusely across the troposphere without a peak-like structure in the ozonesonde profile. In other words, STT events which might have occurred some distance and time away from the location of the ozonesonde profiles may not be readily detected using the high vertical resolution, but infrequent, ozonesonde launches.

We define the ozone peak as the altitude where the perturbation profile is greatest between 2 km from the surface and 0.5 km below the tropopause. The STT event is confirmed if the perturbation profile drops below zero between the ozone peak and the tropopause, as this represents a return to non-enhanced ozone concentrations. Alternatively, the STT event is also confirmed if the OMR between the ozone peak and the tropopause drops below 80 ppb and is at least 20 ppb lower than the OMR at the ozone peak. If neither of these conditions are met, the profile is rejected as a non-event. This final step removes near-tropopause anomalies for which there is insufficient evidence of detachment from the stratosphere. Vertical ozone profiles recorded by ozonesondes are highly dependent on the time of launch (Sprenger et al., 2003), and it cannot be guaranteed that detected ozone enhancements are fully separated from the stratosphere, although this method minimises that risk by removing detected events too near the tropopause.

We estimate the ozone flux into the troposphere associated with each event by integrating the ozone concentration enhancement vertically over the altitude range for which an STT event is identified (i.e. enhancement near the ozone peak over which the perturbation profile is greater than zero). This estimate is conservative because it does not take into account any ozone

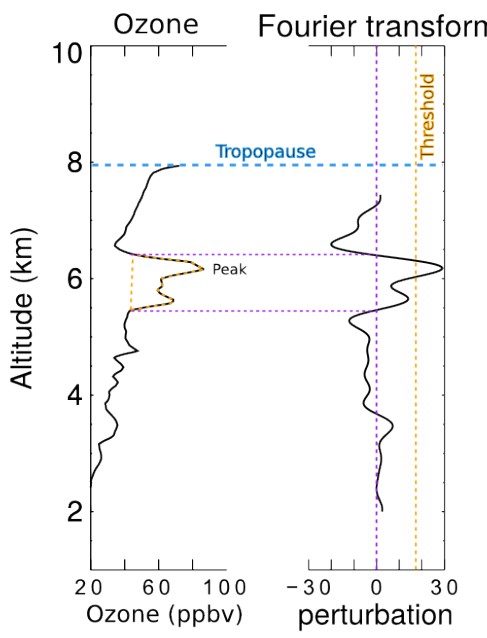

**Figure 4.** An example of the STT identification and flux estimation methods used in this work. The left panel shows an ozone profile from Melbourne on 8 January 2004 from 2 km to the tropopause (blue dashed horizontal line). The right panel shows the perturbation profile created from bandpass filtering of the mixing ratio profile. The STT occurrence threshold calculated from the 95th percentile of all perturbation profiles is shown as the orange dashed line, and the vertical extent of the event is shown with the purple dashed lines (see details in text). The ozone flux associated with the STT event is calculated using the area outlined with the orange dashed line in the left panel.

enhancements outside of the detected peak that may have been caused by the STT, and also ignores any enhanced ozone background amounts from synoptic-scale stratospheric mixing into the troposphere.

Our method differs somewhat from that used by Tang and Prather (2010) to detect STT events from ozonesonde measurements. Their definition is based on subjective analysis of sondes released from 20 stations ranging in latitude from 35° S to 40° N. They identify an STT event if, starting from 5 km altitude, ozone exceeds 80 ppb and then within 3 km decreases by 20 ppb or more to a value less than 120 ppb. Their technique would miss many events due to the lower ozone concentrations found in the cleaner Southern Hemisphere.

### 2.4 Biomass burning influence

The STT detection algorithm described in Sect. 2.3 assumes all ozone enhancements are caused by stratospheric intrusions. In some cases, however, these perturbations may in fact reflect ozone production in lofted smoke plumes. Biomass burning in southern Africa and South America has previously been shown to have a major influence on atmospheric composition in the

vicinity of our measurement sites (Oltmans et al., 2001; Gloudemans et al., 2007; Edwards et al., 2006), particularly from July to December (Pak et al., 2003; Liu et al., 2017). On occasion, smoke plumes from Australian and Indonesian fires can also reach the mid-high southern latitudes, as seen from satellite measurements of carbon monoxide (CO) discussed below.

Large biomass burning events emit substantial quantities of ozone precursors, some of which are capable of being transported over long distances and driving ozone production far from the fire source (Jaffe and Wigder, 2012). Ozone production from biomass burning is complex and affected by photochemistry, fuel nitrogen load, and time since emission, among other factors. While ozone production occurs in some biomass burning plumes, this is not always the case; therefore ozone perturbations detected during transported smoke events may or may not be caused by the plume. For this reason all detected STT events which could be caused by smoke plumes are flagged, following the procedure outlined below. Calculations of seasonality, and ozone flux do not include flagged events, however they are included in summary plots in this work.

Possible biomass burning influence is identified using satellite observations of CO from the AIRS (Atmospheric Infrared Sounder) instrument on board the Aqua satellite (Texeira, 2013). CO is emitted during incomplete combustion and is an effective tracer of long-range transport due to its long lifetime (Edwards, 2003; Edwards et al., 2006). In the Southern Hemisphere, biomass burning is the primary source of CO, making CO a good proxy for fire plumes (e.g. Sinha et al., 2004; Mari et al., 2008). To identify possible biomass burning influence, AIRS vertical column CO is visually inspected for all dates with detected STT events. Smoke plumes are diagnosed over areas with elevated CO columns ($\sim 2 \times 10^{18}$ molecules cm$^{-2}$ or higher), and any sonde-detected STT event that occurs near (within $\sim$150 km of) a smoke plume is flagged. Removal of these detections reduces the yearly estimated ozone flux by $\sim 15\%$ at Macquarie Island and $\sim 20\%$ at Melbourne.

All days with detected STT events were screened, with the exception of one event during which there were no available AIRS data (January 2010). We find that biomass burning may have influenced 27 events over Melbourne and 21 events over Macquarie Island. These events are flagged in the following sections, and are not used in our calculation of total STT flux. All of the flagged events except for two occurred during the SH burning season (July to December). No events at Davis were seen to be influenced by smoke transport.

## 2.5 Classifying synoptic conditions during STT events

Synoptic scale weather patterns are examined using data from the European Centre for Medium-range Weather Forecasts (ECMWF) Interim Reanalysis (ERA-I) (Dee et al., 2011). This is done using the ERA-I data products over the three sites on dates matching the detected STT events. We use the ERA-I 500 hPa data to subjectively classify the events based on their likely meteorological cause, by visually examining each date where an event was detected. During STT occurrence, the upper troposphere is typically characterised by nearby cyclones, cut-off lows, or cold fronts. Over Melbourne and Macquarie Island, we find that frontal and low pressure activity are prevalent during STT events (see Sect. 3). Over Davis, the weather systems are often less clear, however we see a higher portion of probable cut-off lows. The stratospheric polar vortex may create tropopause folds without other sources of upper tropospheric turbulence such as low pressure fronts or cyclones (e.g. Baray et al., 2000; Sprenger et al., 2003; Tyrlis et al., 2014). Cut-off low pressure systems can be seen clearly in synoptic scale weather maps as regions with lowered pressure and cyclonic winds. Low pressure fronts in the higher southern latitudes travel from west to east

**Table 2.** Total number of ozonesonde detected STT events, along with the number of events in each category (see text).

| Site | Events | Cut-offs | Frontals | Misc | Fire |
|---|---|---|---|---|---|
| Davis | 80 | 44 | 19 | 17 | 0 |
| Macquarie Island | 105 | 19 | 31 | 34 | 21 |
| Melbourne | 127 | 28 | 31 | 41 | 27 |

and lower the tropopause height. We examine two cases in detail to illustrate the relationship between synoptic-scale conditions and STT events over Melbourne. These are included in a supplementary document (Fig. S2 and S3) which show an archetypal cut-off low and low pressure front. To detect cut-off low pressure systems we look for cyclonic winds and a detached area of low pressure within $\sim 500$ km of a site on days of event detection. For low pressure fronts we look for low pressure troughs within $\sim 500$ km. Frontal passage is a known cause of STT as stratospheric air descends and streamers of ozone-rich air break off and mix into the troposphere (Sprenger et al., 2003).

## 3 STT event climatologies

Figure 5 shows the seasonal cycles of STT frequency at Davis, Macquarie Island, and Melbourne. Frequency is determined as detected event count divided by total launched ozonesondes for each month. STT events in Figures 5-8 are coloured based on the meteorological classification described in Sect. 2.5, with events classified as either low pressure fronts ("frontal", dark blue), cut-off low pressure systems ("cutoff", teal), or indeterminate ("misc", cyan). Events that may have been influenced by transported smoke plumes (Sect. 2.4) are shown in red. Ozonesonde releases are summarised in Table 1 and detected event counts are summarised in Table 2.

There is an annual cycle in the frequency of STT events (Fig. 5) with a summertime peak at all three sites. This summertime peak is due to a prevalence of summer low-pressure storms and fronts, which increase turbulence and lower the tropopause (Reutter et al., 2015). At Davis, there are more STT detections during winter relative to our other sites, which may be due to the polar vortex and its associated lowered tropopause and increased turbulence. STT events associated with cut-off low pressure systems are more prevalent during summer, while STT events associated with frontal passage occur throughout the year. The high frequency of STT ozone enhancements is comparable to the $> 25\%$ frequencies seen over Turkey and east of the Caspian sea in an ERA-I analysis performed by Tyrlis et al. (2014).

The SH summer maximum we see for STT ozone flux can also be seen in Fig. 16 of Škerlak et al. (2014), which shows seasonal flux over the southern ocean, although this is less clear over Melbourne. This seasonality is not clear in the recent ERA-Interim tropopause fold analysis performed by Škerlak et al. (2015), where a winter maximum of ozone fold frequency ($\sim 0.5\%$ more folds in winter) over Australia can be seen to the north of Melbourne. Their work seems to show slightly higher fold frequencies over Melbourne in summer (Škerlak et al., 2015, Fig. 5), which agrees with our ozonesonde measurements. Their winter maximum is in the subtropics only - from around $20°$ S to $40°$ S, which can be seen as the prevalent feature over Australia in their Fig. 5. Wauben et al. (1998) look at modelled (CTM driven by ECMWF output) and measured ozone

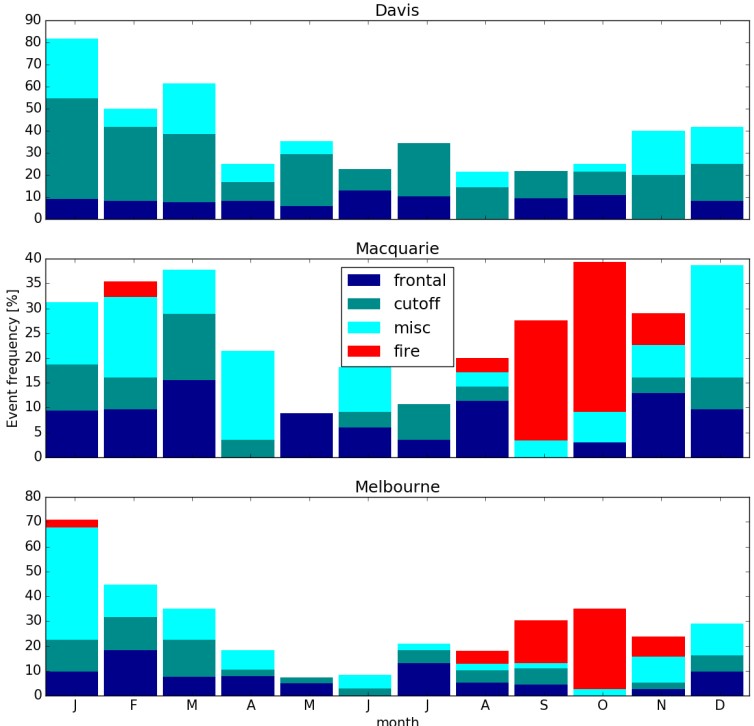

**Figure 5.** Seasonal cycle of STT event frequency at Davis (top), Macquarie Island (middle), and Melbourne (bottom). Events are categorised by associated meteorological conditions as described in the text, with low pressure fronts ("frontal") in dark blue, cut-off low pressure systems ("cutoff") in teal, and indeterminate meteorology ("misc") in cyan. Events that may have been influenced by transported smoke plumes are shown in red (see text for details).

distributions and find more SH ozone in the lower troposphere during austral winter, however they note that the ECMWF fields are uncertain here again due to lack of measurements. Their work shows a generally cleaner lower troposphere in the SH summer but this can not be construed to suggest more or less STT folds in either season. Sprenger et al. (2003) examine modelled STT folds using ECMWF output over March 2000 - April 2001, and show that for this year there is a clear austral winter maximum, again over the 20° S to 40° S band. The winter maximum does not include Melbourne, or the southern ocean, which explains why we see a seasonality not readily evident in these global-scale studies.

The measurement sites are not in the regions which have a clear winter maximum seen in (Fig. 1 Sprenger et al., 2003), and the large scale winter maximum shown by all three studies seems to be dominated by the system in that region. The seasonality of our three sites is not driven by the larger STT system seen over the southern Indian ocean and middle Australia which dominates prior analysis near or over Australia.

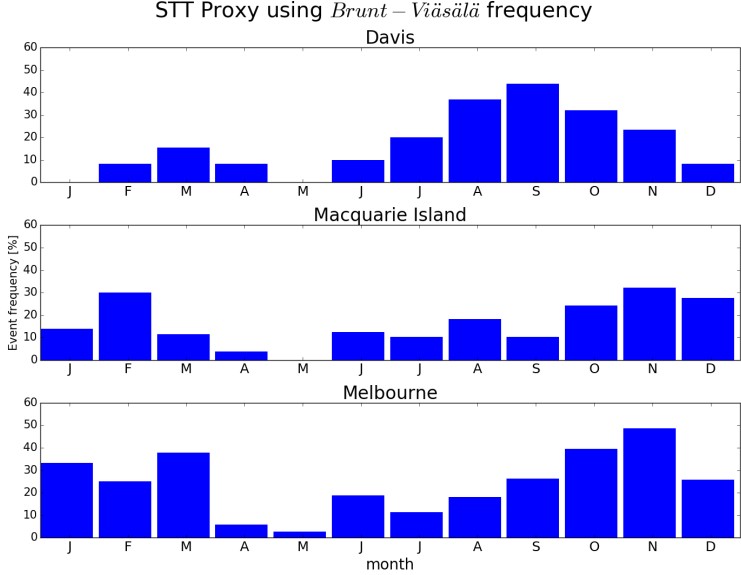

**Figure 6.** Seasonal distribution of STT events using the alternative STT proxy, obtained from consideration of the static stability at the ozone and lapse rate tropopauses, for Davis (2006-2013), Macquarie Island (2004-2013), and Melbourne (2004-2013).

To examine the robustness of the distributions shown in Fig. 5, we developed an alternative assessment of the seasonal occurrence of STT events, with results shown in Fig. 6. Here STT occurrence is evaluated by consideration of the square of the dry Brunt-Väisälä frequency ($N^2$) at the heights of the ozone tropopause ($z_{OT}$) and lapse rate tropopause ($z_{LRT}$) in each ozonesonde profile that has been binned to 100 m resolution. We use $N^2$ to assess atmospheric stability, which is normally distinctly higher in the stratosphere than in the troposphere, and assume that the vertical temperature gradients within the intrusion respond most rapidly to transported heat, which is an additional characteristic of stratospheric air. $N^2$ is evaluated using 250 m resolution data (to smooth variability in the vertical gradient of potential temperature that is due to small temperature fluctuations likely associated with gravity waves). The altitude binning chosen is a compromise between vertical resolution

5    and the level of variability in $N^2$ introduced by temperature gradients associated with perturbations from gravity waves and changes near the lapse rate tropopause, and is the minimum that produces a robust seasonal distribution. We define STT as having taken place if $N^2(z_{OT}) > N^2(z_{LRT})$ and $z_{OT} < z_{LRT}$; in this way the characteristically higher static stability and ozone concentration of stratospheric intrusion is regarded as being retained as it penetrates below the lapse rate tropopause. The seasonal distributions shown for the three stations in Fig. 6 are generally similar to those shown in Fig. 5 (although detected

10    events are less frequent), with the main exception that very few events are identified with the alternative method at Davis in the first half of the year. For our STT proxy, we only detect intrusions where the lowest altitude of the intrusion satisfies the ozone tropopause definition. During summer and autumn, the vertical ozone gradients at Davis are weaker compared with the other seasons, and the detected ozone tropopause tends to lie above the lapse rate tropopause potentially reducing the ability to identify STT events based on the definition of our proxy.

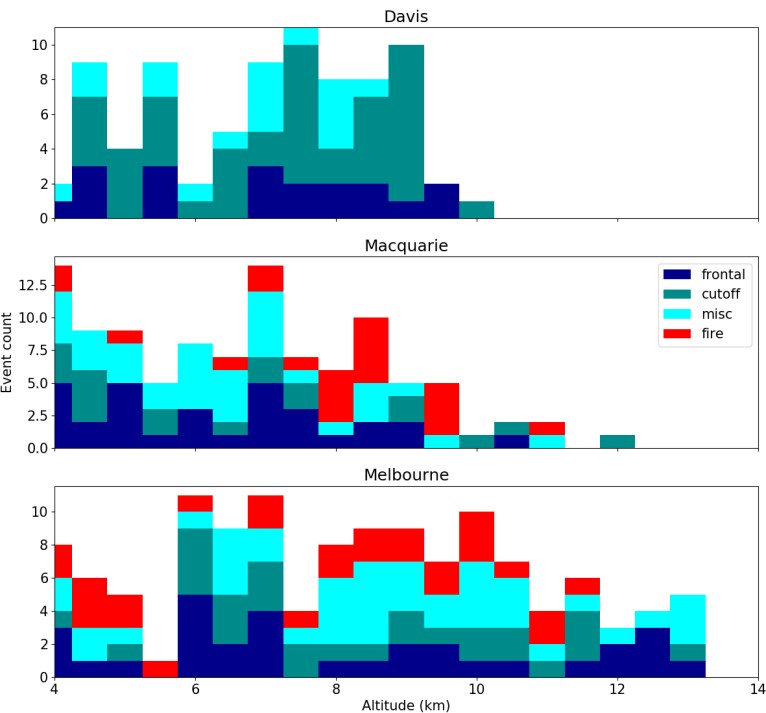

**Figure 7.** The distribution of STT event altitude at Davis (top), Macquarie Island (middle), and Melbourne (bottom), determined as described in the text. Events are coloured as described in Fig. 5.

Figure 7 shows the altitudes of detected events, based on the altitude of peak tropospheric ozone (local maximum ozone within enhancement altitude) in the ozonesonde profile. STT event peaks most commonly occur at 6–11 km above Melbourne and anywhere from 4–9 km at Davis and Macquarie Island. There is no clear relationship between meteorological conditions and event altitude, which may reflect the fact that the ozonesondes observe a snapshot of an event at different stages of its life cycle.

Figure 8 shows the distance from the event peak to the ozone defined tropopause, referred to as event depth. The majority of STT events occur within 2.5 km of the tropopause at Davis and Macquarie Island. Over Melbourne, the event depth is more spread out, with peak ozone enhancement generally occurring up to 6 km below the tropopause. Again, there is no clear relationships between meteorological conditions and event depth.

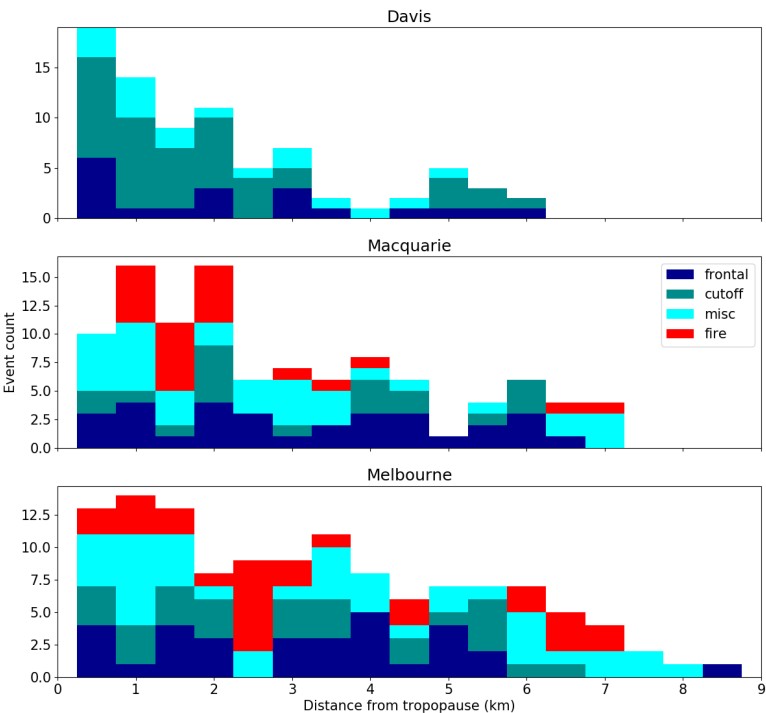

**Figure 8.** The distribution of STT event depth, defined as the distance from the event to the tropopause, at Davis (top), Macquarie Island (middle), and Melbourne (bottom), determined as described in the text. Events are coloured as described in Fig. 5.

## 4  Simulation of ozone columns

Figure 9 compares the time series of tropospheric ozone columns ($\Omega_{O_3}$) in molecules cm$^{-2}$ simulated by GEOS-Chem (red) to the measured tropospheric ozone columns (black). GEOS-Chem outputs ozone density (molecules cm$^{-3}$), and height of each

5  simulated box, as well as which level contains the tropopause, allowing modelled $\Omega_{O_3}$ to be calculated as the product of density and height summed up to the box below the tropopause level. In both observations and model, the maximum ozone column at Melbourne occurs in austral summer, with a minimum in winter, while Macquarie Island and Davis show the opposite seasonality.

GEOS-Chem provides a reasonable simulation of the observed seasonality and magnitude of $\Omega_{O_3}$. Reduced major axis

10  regression of observed versus simulated $\Omega_{O_3}$ gives a line of best fit with slopes of 1.08 for Davis, 0.99 for Macquarie Island, and 1.34 for Melbourne. The model is only partially able to reproduce the variability in the observations, with paired r$^2$ values of 0.38 for Davis, 0.18 for Macquarie Island, and 0.37 for Melbourne. Much of the variability is driven by the seasonal cycle,

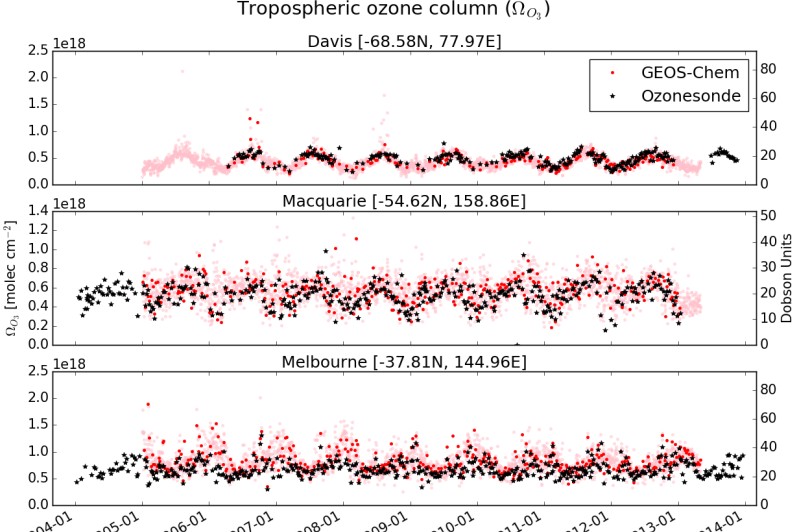

**Figure 9.** Comparison between observed (black) and simulated (pink, red) tropospheric ozone columns ($\Omega_{O3}$, in molecules cm$^{-2}$) from 1 January 2004 to 30 April 2013. For the model, daily output is shown in pink, while output from days with ozonesonde measurements are shown in red. For each site, the model has been sampled in the relevant grid square.

and after removing this effect (by subtracting the multi-year monthly means), the r$^2$ values decrease to 0.07, 0.11, and 0.30 respectively, although the slope improves at Melbourne to 1.08.

5     Figure 10 shows the observed and simulated ozone profiles at all sites, averaged seasonally. The model generally underestimates ozone in the lower troposphere (up to 6 km) over Davis, although this bias is less pronounced during summer. Over Melbourne, ozone in the lower troposphere is well represented, but the model overestimates ozone from around 4 km to the tropopause. Over Macquarie Island we see model overestimation of ozone above 4 km, as well as underestimated ozone in the lower troposphere, suggesting that this region is influenced by processes seen at both of our other sites. Also shown is the mean

10     tropopause height simulated by the model (horizontal dashed red line), which is always higher than the observed average, although this difference is not statistically significant. The effect of local pollution over Melbourne during austral summer (DJF) can be seen from the increased mean mixing ratios and enhanced variance near the surface. The gradient of the O$_3$ profiles is steeper in the measurements than the model, at all sites during all seasons. Recently Hu et al. (2017) examined GEOS-Chem ozone simulations and saw a similar overestimation of upper troposphere ozone in the mid southern latitudes when running

5     with GEOS5 meteorological fields.

    Figure 11 compares modeled (red) and observed (black) ozone profiles on three example days when STT events were detected using the ozonesondes. The figures show the profile for each site with the closest (qualitative) match between model and observations. The resolution (both vertical and horizontal) of GEOS-Chem in the upper troposphere is too low to consistently

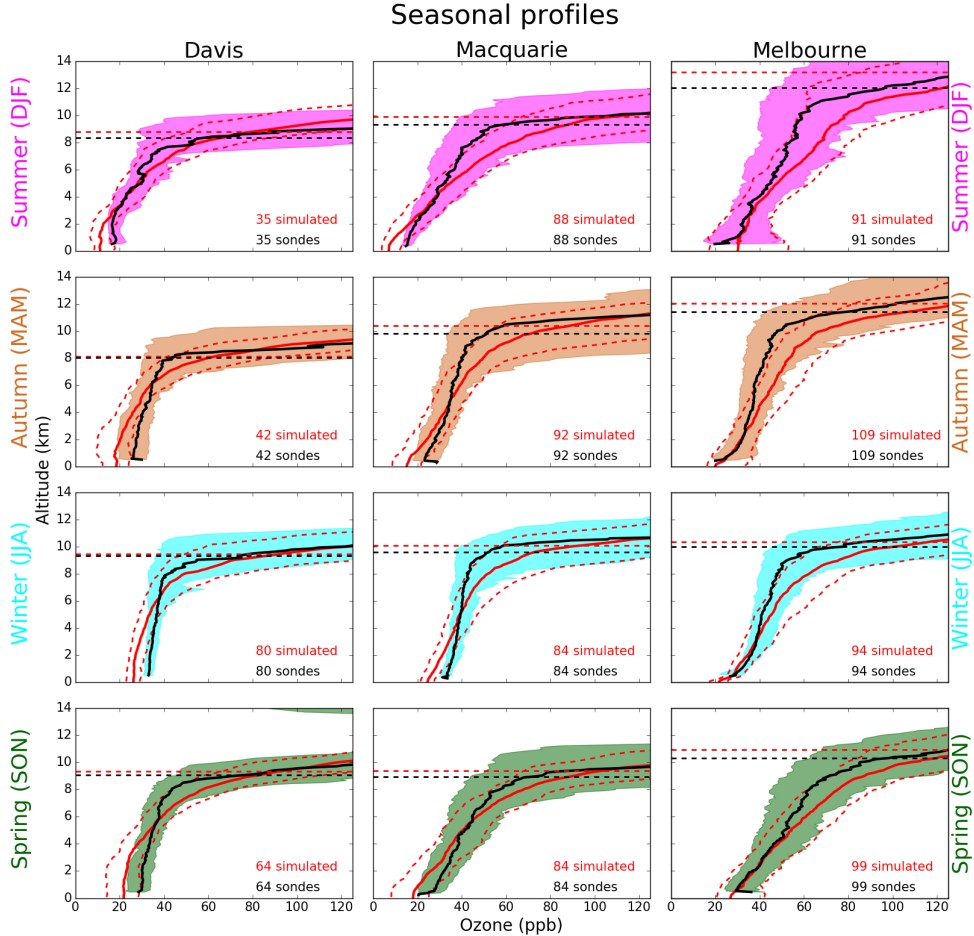

**Figure 10.** Observed and simulated tropospheric ozone profiles over Davis, Macquarie Island, and Melbourne, averaged seasonally. Model medians (2005-2013 average) are shown as red solid lines, with red dashed lines showing the 10th and 90th percentiles. Ozonesonde medians (over each season, for all years) are shown as black solid lines, with coloured shaded areas showing the 10th and 90th percentiles. The horizontal dashed lines show the median tropopause heights from the model (red) and the observations (black).

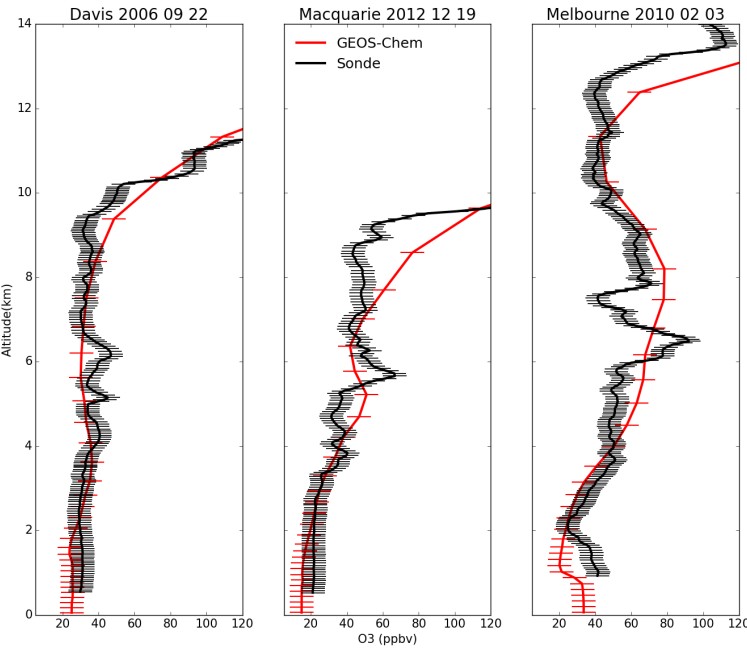

**Figure 11.** Example comparisons of ozone profiles from ozonesondes (black) and GEOS-Chem (red) from three different dates during which STT events were detected from the measurements. The dates were picked based on subjective visual analysis. The examples show the best match between model and observations for each site. GEOS-Chem and ozonesonde pressure levels are marked with red and black dashes respectively.

allow detection of STTs, although in a few cases (e.g., Melbourne in Fig. 11) it appears that the event was large enough to be visible in the model output.

# 5 Stratosphere-to-troposphere ozone flux from STT events

## 5.1 Method

We quantify the mean stratosphere-to-troposphere ozone flux due to STT events at each site based on the integrated ozone amount associated with each STT event (see Sect. 2.3). Events that may have been influenced by transported biomass burning are excluded from this calculation. Our estimate provides a preliminary estimate of how much ozone is transported from the stratosphere by the events detected by our method. The estimate is conservative for several reasons: it ignores secondary ozone peaks which may also be transported from the stratosphere, it ignores potential ozone enhancements which may have dispersed

and increased the local background mixing ratio, and any influence from STT events nearby which may also increase the local background ozone.

Observed tropospheric columns are calculated from the ozonesondes by calculating the ozone number density (molecules cm$^{-3}$) using measured ozone partial pressure ($P_{O_3}$) and integrating vertically up to the tropopause:

$$\Omega_{O_3} = \int\limits_0^{TP} \frac{P_{O_3}(z)}{k_B \times T(z)} \mathrm{d}z$$

where $z$ is the altitude (GPH), $TP$ is the altitude at the tropopause, $T$ is the temperature, and $k_B$ is the Boltzmann constant.

Three regions are used to examine possible STT flux over a larger area using modeled tropospheric ozone concentrations. The regions are shown in Fig. 1. The regions are centred at each site, plus or minus ten degrees latitude, and plus or minus 25, 16, and 11 degrees longitude for Davis, Macquarie Island, and Melbourne respectively. These boundaries approximate a rectangle centred at each site with $\sim 2000$ km side lengths, covering $\sim 4.4$, $4.6$, and $4.8$ million square km, for Davis, Macquarie Island, and Melbourne respectively.

To determine the ozone column attributable to STT, we determine monthly averaged STT impact ($I$; fraction of tropospheric ozone sourced from the stratosphere as shown above) and the monthly mean tropospheric ozone column (from the GEOS-Chem multi-year mean, $\Omega_{O_3}$) over the regions described above. This can be expressed simply as the STT flux per event (flux$_i$ in each month: flux$_i = \Omega_{O_3} \times I$. Next we determine how many events are occurring per month by assuming only one event can occur at one time, and that no event is measured twice. These assumptions allow a simple estimate of events per month from the relatively sparse dataset and should hold true as long as our regions of extrapolation are not too large. The ($P$)robability of any sonde launch detecting an event is calculated as the fraction of ozonesonde releases for which an STT event was detected, calculated for each month. If we assume events last N days, we find how many events per month by multiplying the days in a month by $P$ and dividing by the assumed event lifetime. For example if we expect to see an event 25% of the time in a month, and events last one day, we expect one event every four days ($\sim 7.5$ events in that month) whereas if we expect events to last a week then we would expect $\sim$one event in that month. This leads us to multiply our flux$_i$ by $P$, and then by the term $M$ ($M = \frac{\text{days per month}}{N}$) determined by our assumed event lifetime in order to determine monthly STT ozone flux.

The longevity of ozone events is very difficult to determine, and we have chosen 2 days as a representative number based on several examples in Lin et al. (2012) where intrusions were seen to last from 1-3 days (occasionally longer) and an analysis of one large event by Cooper et al. (2004) showing that most of the ozone had dispersed after 48 hours. Worth noting is the recent work of Trickl et al. (2014), where intrusions are detected $> 2$ days and thousands of kilometres away from their initial descent into the troposphere over Greenland or the Arctic. In those regions with high wind shear, mixing appears to be slower, which allows ozone intrusions to be transported further without dissipating into the troposphere. Relative uncertainty in our $M$ term is set to 50%, as we assume these synoptic events to generally last from 1-3 days.

**Table 3.** Seasonal STT ozone contribution in the regions near each site, in kg km$^{-2}$ month$^{-1}$. In parentheses are the relative uncertainties.

| Region | DJF | MAM | JJA | SON |
|---|---|---|---|---|
| Davis | 54.5 (102%) | 47.7 ( 97%) | 30.7 (114%) | 18.8 (127%) |
| Macquarie Island | 61.3 ( 85%) | 70.7 ( 91%) | 17.9 (139%) | 7.7 (229%) |
| Melbourne | 96.7 (103%) | 88.6 ( 89%) | 26.7 (102%) | 21.4 (109%) |

## 5.2 Results

The top panel of Fig. 12 shows the STT ozone enhancements, based on a vertical integration of the ozone above baseline levels for each ozonesonde where an event was detected. The area considered to be 'enhanced' ozone is outlined with yellow dashes on the left panel of Fig. 4. We find that the mean ozone flux associated with STT events is $\sim 0.5$–$2.0 \times 10^{16}$ molecules cm$^{-2}$. The bottom panel shows the mean fraction of total tropospheric column ozone (calculated from ozonesonde profiles) attributed to stratospheric ozone intrusions at each site for days when an STT event occurred. First the tropospheric ozone column is calculated, then the enhanced ozone column amount is used to determine the relative increase. At all sites, the mean fraction of tropospheric ozone attributed to STT events is $\sim 1.0$–$3.5\%$. On three seperate days over Macquarie and Melbourne, this value exceeds 10%.

The upper panels in figures 13-15 show the factors $I$, $P$, and $\Omega_{O_3}$ which are used along with the assumed event lifetime to estimate the STT flux. The tropospheric ozone and area of our region is calculated using the output and surface area from GEOS-Chem over our three regions. The lower panel of these figures show the results of the calculation when we choose two days for our flux estimation, with dotted lines showing the range of flux calculated if we assume events last from one day to one week. The seasonal cycle of ozone flux is mostly driven by the $P$ term, which peaks in the SH summer over all three sites. Total uncertainty (shaded) is on the order of $100\%$ (see Sect. 6.2). We calculate the annual amount based on the sum of the monthly values. The regions over Davis, Macquarie Island, and Melbourne have estimated STT ozone contributions of $\sim 5.7 \times 10^{17}$, $\sim 5.7 \times 10^{17}$, and $\sim 8.7 \times 10^{17}$ molecules cm$^{-2}$ a$^{-1}$ respectively, or equivalently $\sim 2.0$, 2.1, and 3.3 Tg a$^{-1}$.

## 5.3 Comparison to literature

Škerlak et al. (2014) show an estimate of roughly 40 to 150 kg km$^{-2}$ month$^{-1}$ in these regions, over all seasons (see Fig. 16, 17 in their publication) while we estimate from 0 to 180 kg km$^{-2}$ month$^{-1}$ STT impact, following a seasonal cycle with the maximum in austral summer. We estimate higher maximum flux over Melbourne, (178, and 150 kg km$^{-2}$ month$^{-1}$ in January and February) than in either Davis (89 kg km$^{-2}$ month$^{-1}$ in March) or Macquarie Island (68 kg km$^{-2}$ month$^{-1}$ in January). Our calculated seasonal contributions, along with total uncertainty are shown in Table 3.

This result disagrees with several other studies which have found STT ozone fluxes in the SH extra-tropics are largest from autumn or winter to early spring. Roelofs and Lelieveld (1997) use a model with a stratospheric ozone tracer to estimate STT impacts. They see higher SH tropospheric ozone concentrations, as well as STT flux, in the SH winter. Our model also shows ozone column amounts peaking in winter, however flux is maximised in summer due to our detected event frequencies. Elbern

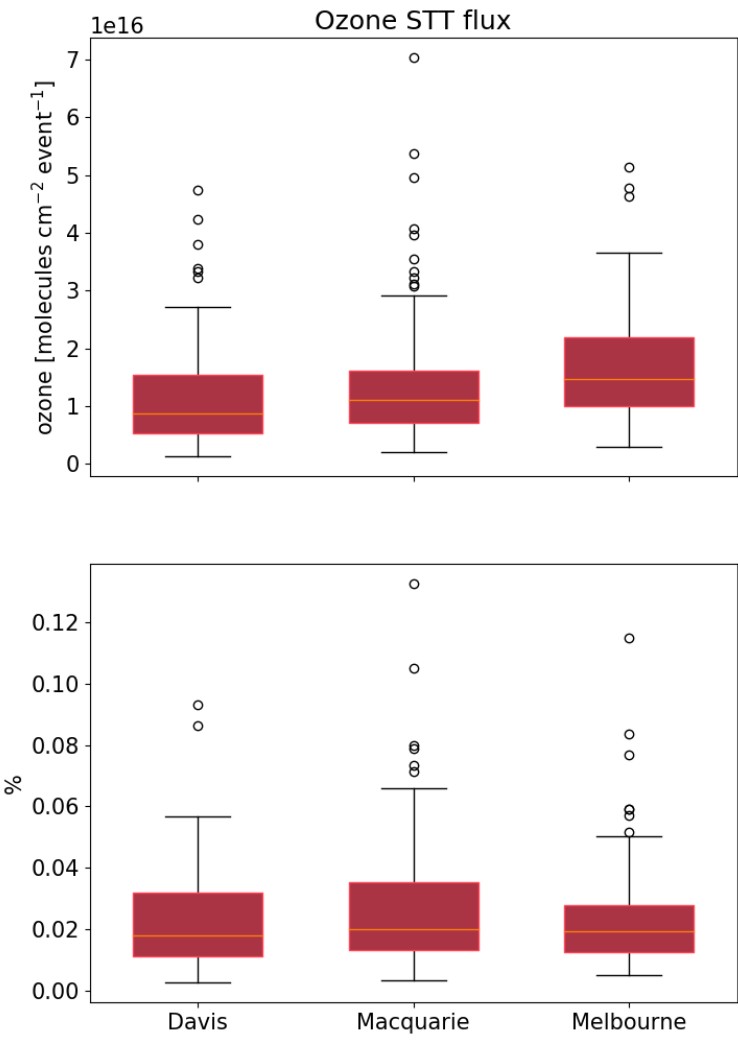

**Figure 12.** Top panel: tropospheric ozone attributed to STT events. Bottom panel: percent of total tropospheric column ozone attributed to STT events. Boxes show the inter-quartile range (IQR), with the centre line being the median, whiskers show the minimum and maximum, circles show values which lie more than 1.5 IQR from the median. Values calculated from ozonesonde measurements as described in the text.

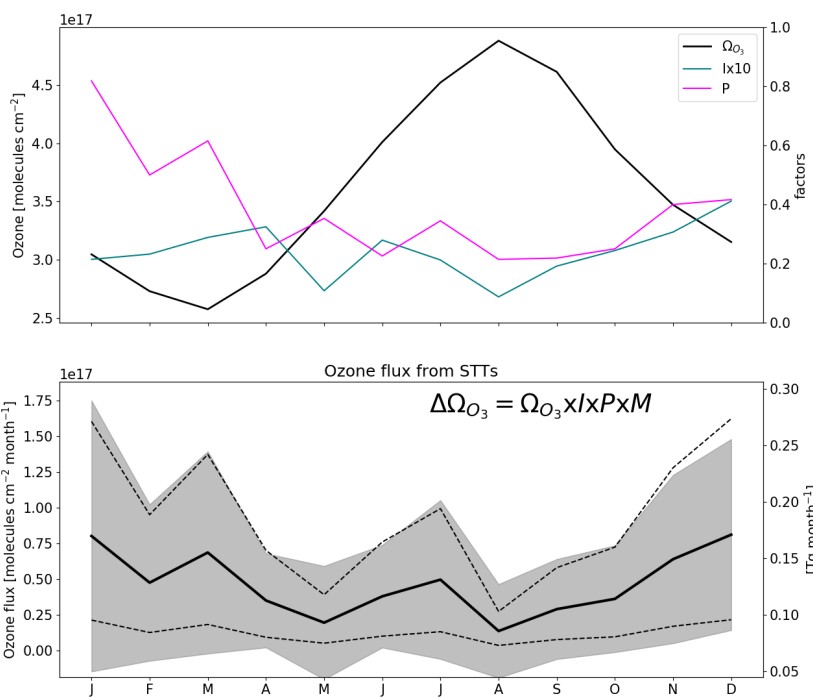

**Figure 13.** (Top) Tropospheric ozone, ($I$)mpact per event, and ($P$)robability of event detection per sonde launch, averaged over the region above Davis. The tropospheric ozone column $\Omega_{O_3}$ (black, left axis) is from GEOS-Chem, while the STT probability $P$(magenta, right axis) and impact $I$ (teal, right axis) are from the ozonesonde measurements. The STT impact is multiplied by ten to better show the seasonality. (Bottom) Estimated contribution of STT to tropospheric ozone columns over the region, with uncertainty (shaded area) estimated as outlined in Sect. 6. The black line shows STT ozone flux if event lifetime is assumed to be two days, with dashed lines showing the range of flux estimation if we assumed events lasted from one day to one week.

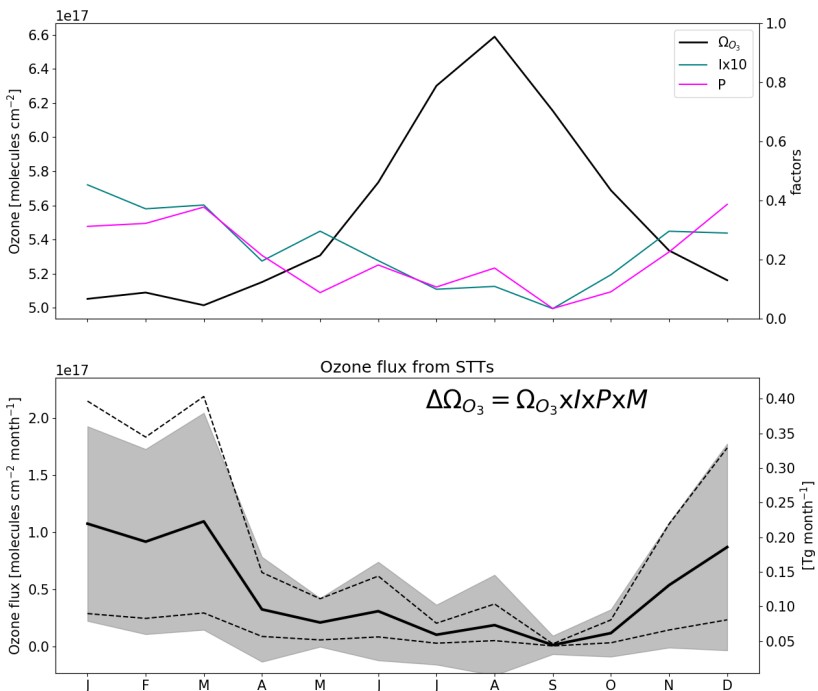

**Figure 14.** As described in 13, for the region containing Macquarie Island.

et al. (1998) examine STT using ECMWF data for prior to 1996, using PV and Q-vectors to determine STT frequency and strength, and suggest fewer fold events in the SH occur from December to February. Olsen (2003) used PV and winds from the GEOS reanalysis combined with ozone measurements from the TOMS satellite to estimate that the ozone flux between $30°$ S and $60°$ S is 210 Tg yr$^{-1}$, with the maximum occurring over SH winter. Liu et al. (2017) model the upper tropospheric ozone and its source (emissions/lightning/stratospheric) over the Atlantic ocean between $30°$ S and $45°$ S, and suggest that most of this is transported from the stratosphere from March to September, which is when the subtropical jet system is strongest.

The disagreements largely reflect the difference between point source based estimates and zonally averaged estimates, as the meteorological behaviour at our three sites is not the same as the system that dominates the southern hemisphere in general. As detailed in Sect. 3, the maximum STT influx which occurs during SH winter is almost entirely due to the dominant STT system which occurs annually over the southern Indian ocean and middle of Australia. It is difficult to compare remote ozonesonde datasets with area averaged models or re-analyses based on non-co-located measurements (such as ERA).

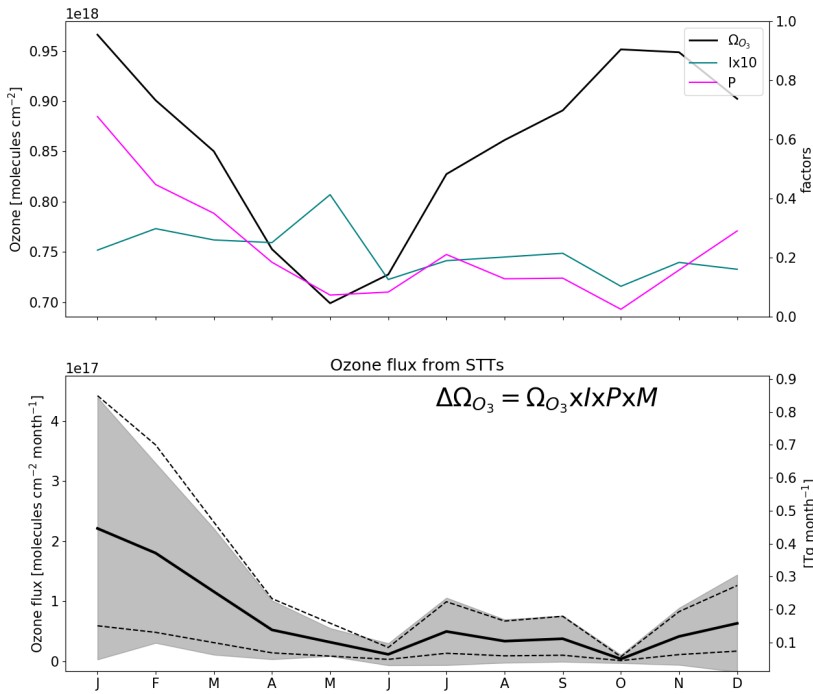

**Figure 15.** As described in 13, for the region containing Melbourne.

# 6 Sensitivities and limitations

## 6.1 Event detection

Our method uses several subjectively-defined quantities in the process of STT event detection. Here we briefly discuss these quantities and the sensitivity of the method to each. Using the algorithm discussed in Sect. 2.3, we detect 80 events at Davis, 105 (21 fire influenced) events at Macquarie Island, and 127 (27 fire influenced) events at Melbourne.

The cut-off threshold (defined separately for each site) is determined from the 95th percentile of the ozone perturbation profiles between 2 km above the earth's surface and 1 km below the tropopause. We use the 95th percentile because at this point the filter locates clear events with fewer than 5% obvious false positive detections. Event detection is sensitive to this choice; for example, using the 96th, and 97th percentile instead decreased detected events by 2, 9 (2,10%) at Davis, 13, 31 (11, 28%) at Macquarie Island, and 8, 24 (6, 18%) at Melbourne. Event detection is therefore also sensitive to the range over which the percentile is calculated. This range was chosen to remove anomalous edge effects of the Fourier bandpass filter and to discount the highly variable ozone concentration which occurs near the tropopause.

Ozone enhancements are only considered STT events if they occur from 4 km altitude up to 500 m below the tropopause.
This range removes possible ground pollution and events not sufficiently separated from the stratosphere, while still capturing many well-defined events that occur within 1 km of the tropopause. An example of a well-defined event that occurs within 1 km of the tropopause is shown in the supplementary (Fig. S2). However, STT events which reach below 4 km are physically possible and we may have some false negative detections due to the altitude restricted detections.

## 6.2 Flux calculations

Flux is calculated as $I \times P \times M \times \Omega_{O_3}$, with each term calculated as described in Sect. 5.1. The uncertainty is determined using the standard deviation of the product, with variance calculated using the variance of a product formula, assuming that each of our terms is independent:

$$var(\Pi_i X_i) = \Pi_i(var(X_i) + E(X_i)^2) - (\Pi_i E(X_i))^2$$

The standard deviations for the $I$ and $\Omega_{O_3}$ terms are calculated over the entire dataset. These terms are considered to be homoskedastic (unchanging variance over time). Uncertainty in assumed event lifetime is set at 50%, as we believe it is reasonable to expect events to last 1-3 days. $P$ is the probability of any ozonesonde detecting an event, and is assumed to be constant (for any month). The overall uncertainty as a percentage is shown in parentheses in Table 3, these values are on the order of 100%, largely due to relative uncertainty in the $I$ factor which ranges from 50-120% for each month.

Small changes in the region don't have a large affect on the per area flux calculations: increasing or decreasing the regions by $1°$ on each side ($\sim 10\%$ change in area) change the resulting flux by $\sim 1\%$. However due to the large portion of winter STT events being flagged due to potential smoke plume influence, a significant change in the yearly flux is seen when we don't remove these events. Without removing smoke flagged events we see an increase in estimated yearly flux of $\sim 1.1, 2.1 \times 10^{17}$ molecules cm$^{-2}$ yr$^{-1}$ (which is a change of $\sim 15, 20\%$), over Macquarie Island and Melbourne respectively.

Considering the $I$ factor, as discussed in here and in Sect. 6, there are several uncertainties in our method that are likely to lead to a low bias, such as the conservative estimate of flux within each event. Although there is little available data on SH ozone events for us to compare against, consider Terao et al. (2008), who estimated that up to 30–40% of the ozone at 500 hPa was transported from the stratosphere, in the northern hemisphere.

Our STT event impact estimates have some sensitivity to our biomass burning filter: including smoke-influenced days increases the mean per area flux by 15-20%. Although events which are detected near fire smoke plumes are removed, some portion of these could be actual STTs. The change in our P parameter when we include potentially smoke influenced events leads to a yearly estimated STT of $11 \times 10^{17}$ molecules cm$^{-2}$ yr$^{-1}$ over Melbourne, which suggests that up to $2.1 \times 10^{17}$ molecules cm$^{-2}$ yr$^{-1}$ ozone enhancement could be caused by smoke plume transported precursors. This is a potential area for improvement, as a better method of determining smoke influenced columns would improve confidence in our estimate.

Other possibly important uncertainties in our calculation of STT flux which we don't cover are listed here. Filtering events which occur within 500 m of the tropopause may also lead to more false negatives. This could also cause lower impact estimates due to only measuring ozone enhancements which have descended and potentially slightly dissipated. On the other hand we

have no measure of how often the detached ozone intrusion reascends into the stratosphere, which would lead to a reduced stratospheric impact. The estimated tropospheric ozone columns modelled by GEOS-Chem may be biased, for instance Hu et al. (2017) suggest that in general GEOS-Chem (with GEOS-5 met. fields) underestimates STT, with $\sim 360$ Tg a$^{-1}$ simulated globally, compared to $\sim 550$ Tg a$^{-1}$ observationally constrained. Transport uncertainty is very difficult to estimate with the disparate point measurements; it's possible that detected events are (at least partially) advected out of the analysis regions, which would mean we overestimate the influx into the region, and it is also possible that we are influenced by STT events outside the regions of analysis. Uncertainty in event longevity is set to 50%, however this implies a very simplistic model of event lifetimes. A great deal of work could be done to properly model the regional event lifetimes, however this is beyond the scope of our work.

Uncertainties in STT ozone flux detection are ($\sim 100\%$), and could be directly improved with larger or longer datasets. Possible parameterisations and an improved model of event lifetime could also improve the confidence in our estimate of event impacts, as well as allowing fewer assumptions.

# 7 Conclusions

Stratosphere-to-troposphere transport (STT) can be a major source of ozone to the remote free troposphere, but the occurrence and influence of STT events remains poorly quantified in the southern extra-tropics. Ozonesonde observations in the SH provide a satellite-independent quantification of the frequency of STT events, as well as an estimate of their impact and source. Using almost ten years of ozonesonde profiles over the southern high latitudes, we have quantified the frequency, seasonality, and altitude distributions of STT events in the SH extra-tropics. By combining this information with ozone column estimates from a global chemical transport model, we provided a first, conservative estimate of the influence of STT events on tropospheric ozone over the Southern Ocean.

Our method involved applying a bandpass filter to the measured ozone profiles to determine STT event occurrence and strength. The filter removed seasonal influences and allowed clear detection of ozone-enhanced tongues of air in the tropo-sphere. By setting empirically-derived thresholds, this method clearly distinguished tropospheric ozone enhancements that are separated from the stratosphere. Our method is sensitive to various parameters involved in the calculation; however, for our sites we saw few false positive detections of STT events.

Detected STT events at three sites spanning the SH extra-tropics (38°S, 55°S, and 69°S) showed a distinct seasonal cycle. All three sites displayed a summer (DJF) maximum and an autumn to winter (AMJJA) minimum, although the seasonal amplitude was less apparent at the Antarctic site (Davis) as events were also detected regularly in winter and spring (likely due to polar jet stream-caused turbulence). Analysis of ERA-Interim reanalysis data suggested the majority of events were caused by turbulent weather in the upper troposphere due to low pressure fronts, followed by cut-off low pressure systems. Comparison of ozonesonde-measured ozone profiles against those simulated by the GEOS-Chem global chemical transport model showed the model is able to reproduce seasonal features but does not have sufficient vertical resolution to distinguish STT events.

By combining the simulated tropospheric column ozone from GEOS-Chem with ozonesonde-derived STT estimates, we
provide a first estimate of the total contribution of STT events to tropospheric ozone in these southern extra-tropical regions.
We estimate that the ozone enhancement due to STT events near our three sites ranges from 300-570 kg km$^{-2}$ a$^{-1}$, with
seasonal maximum in SH summer.

Estimating STT flux using ozonesonde data alone remains challenging; however, the very high vertical resolution provided
by ozonesondes means they are capable of detecting STT events that models, re-analyses, and satellites may not. Further work
is needed to more accurately translate these ozonesonde measurements into STT ozone fluxes, particularly in the SH where data
are sparse and STT is likely to be a major contributor to upper tropsopheric ozone in some regions. More frequent ozonesonde
releases at SH sites could facilitate development of better STT flux estimates for this region.

*Author contributions.* JWG wrote the algorithms, ran the GEOS-Chem simulations, performed the analysis and led the writing of the paper
under the supervision and guidance of SPA, RS, and JAF. AK contributed the Davis ozonesonde data and performed the analysis of the
alternate STT proxy. All authors contributed to editing and revising the manuscript.

*Competing interests.* The authors declare that they have no conflict of interest.

*Data availability.* All GEOS-Chem model output and the ozonesonde observational data are available from the authors upon request.

*Acknowledgements.* We thank Dr. Sandy Burden for help clarifying some of the uncertainties involved in methods within this work. We
also thank Dr. Clare Paton-Walsh, who identified the need to account for smoke-influenced events, and provided discussions on how to
go about doing such. Ozonesonde data comes from the World Ozone and Ultraviolet Data Centre (WOUDC). The ERA-Interim data were
downloaded from the ECMWF website following registration. This research was undertaken with the assistance of resources provided at the
NCI National Facility systems at the Australian National University through the National Computational Merit Allocation Scheme supported
by the Australian Government. This work was supported through funding by the Australian Government's Australian Antarctic science
grant program (FoRCES 4012), the Australian Research Council's Centre of Excellence for Climate System Science (CE110001028), the
Commonwealth Department of the Environment ozone summer scholar program. This research is supported by an Australian Government
Research Training Program (RTP) Scholarship.

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
