# Peer review of "Stratospheric ozone intrusion events and their impacts on tropospheric ozone in the Southern Hemisphere"

_Atmospheric Chemistry and Physics, 2016_

## Short Comment (SC1) · 19 Jan 2017

1. Lines 20-22: Regarding the influence of STT on US surface ozone air quality, please consider citing the following paper:

Meiyun Lin, A.M. Fiore, L.W. Horowitz, A.O. Langford, S. J. Oltmans, D. Tarasick, H.E. Reider (2015): Climate variability modulates western US ozone air quality in spring via deep stratospheric intrusions, Nature Communications, 6, 7105, doi:10.1038/ncomms8105

http://www.nature.com/articles/ncomms8105

The earlier papers the authors cited do not actually talk about the influence of STT on surface air quality.

[Figure]

2. Line 22-23: "In the western US, for example, STT events have been shown to contribute up to 30% of surface ozone in spring (Lin et al., 2012)."

Consider rephrasing this sentence to:

In the western US, for example, deep STT events during spring can add 20 to 40 ppbv of ozone to the ground-level ozone concentration, which can provide over half the ozone needed to exceed the standard set by the U.S. Environmental Protection Agency (Lin et al., 2012; Lin et al., 2015).

3. Add "in the Southern Hemisphere" to the title of this paper?

---

## Short Comment (SC2) · 25 Jan 2017

We appreciate the suggestions and will include them in the final revised version.

---

## Referee Comment (RC1) · Anonymous Referee #1 · 6 Feb 2017

The current study presents a method to identify stratosphere-to-troposphere transport (STT) events and estimate the associated ozone flux to the troposphere, based on ozonesonde profiles from three sites located in the Southern Hemisphere extratropics. Subsequently, the seasonality of STT events is determined, as well as the favorable synoptic conditions. Based on the stratospheric contribution to tropospheric ozone column estimated from the ozonesondes, the GEOS-Chem simulated tropospheric ozone columns are extrapolated to assess the stratospheric contribution over the Southern Ocean region. As the STT is of great importance for the tropospheric ozone budget and variability, and the number of relevant studies (both observational and modeling) for the examined region is limited, I find the topic of the paper within the scope of ACP. On the other side, there are several issues that need to be addressed before consideration for publication in ACP.

[Figure]

Major Comments:

1) Calculating the 99th percentile from the perturbation profiles over that layer (2 to 1 Km below the tropopause) is a fairly strict criterion. Wouldn't this threshold choice avoid the selection of deeper stratospheric intrusion events as "STT events"? Have you consider modifying this criterion, and include others (e.g. significant negative O3-relative humidity correlation values above a threshold) to minimize false STT detection?

2) The seasonality of STT events presented in Fig. 7 is not in line with the findings of Škerlak et al. (2015) for the examined region. How are your results (STT seasonality) compared with other modeling studies (Elbern et al., 1998; Sprenger et al., 2003)? Is there any evidence from other studies that STT frequency over the examined region exhibits a maximum during the austral summer (DJF) and not during the austral winter (JJA) when the jet stream is strongest over the broader region? Have you tried to detect STT events from the model results? I guess this is strongly depended to the vertical resolution of the model, but it would be very interesting to see how the observed and modeled STT seasonalities are compared.

3) To my understanding, using the seasonality of STT events from the three sites to extrapolate model results over the Southern Ocean region is a quite simplified and coarse approach, especially when considering the previous comment.

4) Overall, the presentation of the results can be further improved (please check my suggestions further below), as well as the writing of the manuscript.

Comments:

Škerlak et al. (2014) presented an STE climatology using the ERA-Interim data. This study is important not only for the introduction, as it describes the STT climatology for the SH, but for intercomparison of the results also. Similar climatologies can be found in the modeling studies of Roelofs and Lelieveld (1997) and James et al. (2003). Recently, Akritidis et al. (2016) explored the impact of stratospheric intrusions on tropo-

spheric ozone and the associated stratospheric contribution over the eastern Mediterranean and the Middle East region, a task that is relevant with some of the purposes of this study.

Page 4, lines 3-4: Since the study is based on the ozonesondes launched from the three sites, it is important to present the location of the sites.

Page 4, line 22: "Figure 1 shows the monthly mean tropopause altitudes at ..", while in Fig. 1 caption is stated "Multi-year monthly median tropopause altitude ..". Is it the mean or the median? Please modify accordingly.

Page 5, Figure 1: a) The shadings used to describe the 10th and 90th percentiles are rather confusing. I suggest you replace the shadings with dashed lines (same color as the solid lines). b) Increase the range of the vertical axis to show the 10th percentile value for February. c) Is it the case that tropopause drops below 4 km (10th percentile) over Davis? What is the minimum tropopause height value over Davis during February?

Page 5, lines 5-6: "This seasonality at the high latitude sites is driven by a decrease in photochemical destruction under the reduced radiation conditions around polar night." Please include a reference or information about the NOx levels at these sites (if available) to justify this statement.

Page 6, line 14: It is important to know the vertical resolution of the GEOS-Chem model near the tropopause (although it can partially be seen from Fig. 13), as it is important for the tropopause height detection and the tropospheric ozone column calculations from the model results.

Page 7, lines 22-23: "The interpolated profiles. . .high frequency perturbations)." This is a rather brief description of the procedure. A more detailed description including a reference (if available) for the FT application would be necessary.

Page 7, lines 27-28: "We next use all the perturbation profiles at each site to calculate the 99th percentile perturbation value for the site". How exactly is this cut-off threshold

calculated? In Section 2.5, Page 9, the authors state that is calculated "between 2 km and 1 km below the tropopause". This information should be provided earlier in the manuscript, at the point that the 99th percentile threshold is initially mentioned (Section 2.3).

Page 8, Figure 3: Why the two panels have different units? Are the ozone units of the left panel "1e+12 molecules cm-3"? Please change accordingly the Figure and the Figure caption. mixing ratio -> number density

Page 9, lines 1-2: "For this reason all detected STT events found near smoke plumes are flagged". How is "near" defined?

In my opinion, Figures 4, 5 and 6 are more supportive-descriptive without adding anything new. Therefore, I suggest including them as a supplement. Moreover, Figures 5 and 6 can be merged into one.

Page 11, line 17: "We use the ERA-I 500 hPa data to subjectively classify the events based on their likely meteorological cause." Do the authors classify the events by visual inspection of the 500 hPa maps for every STT event date?

Page 11, lines 20-21: "The stratospheric polar vortex may create ozone folds without other sources of upper tropospheric turbulence". Please include a reference for the above statement.

Page 14, lines 16-20: "The seasonal distributions ... first half of the year". To my understanding Fig. 7 and Fig. 8 are not quite similar. Moreover, comparing Fig. 8 with Fig. 7 where fire influences are also included is somehow unfair. The fact that ozonesondes are launched monthly at Davis from December to June is also the case for Fig. 7, where high STT frequencies are found for the respective period.

Page 16: How is the modeled tropospheric column ozone calculated? How is the tropopause defined in the GEOS-Chem results?

Page 17, lines 3-4: "Over Melbourne, ozone in the lower troposphere is well represented, but the model overestimates ozone from around 4 km to the tropopause". This is also seen for Macquarie and should be added to the discussion.

Page 19: "Figure 14 shows the mean fraction of total tropospheric column ozone (calculated from ozonesonde profiles) attributed to stratospheric ozone intrusions at each site, averaged over days when an STT event occurred." Please explain in more detail how is this fraction calculated.

Page 19: "to the entire Southern Ocean region, defined here as 35_ S-75_ S to encompass". What is the longitudinal range?

Page 20: Fig. 14 and Fig.15 can be merged into one.

Page 22: "If we we assume a fractional ozone impact due to each event STT event of I=35% based on their results". The 30-40% stratospheric contribution found by Terao et al. (2008) is seen only during spring and at 500 hPa. Therefore, assuming a 35% stratospheric contribution to the tropospheric column ozone seems a bit arbitrary.

Minor comments:

Page 1, line 4: seasonality -> seasonality of STT events

Page 1, line 9: 2.5 km, 3 km -> 2.5 km and 3 km

Page 1, line 14: these -> which

Page 2, line 2: .Despite lingering -> . Despite the lingering

Page 2, line 29: found STT -> found that STT

Page 2, line 31: challenging to accurately represent, and better model resolution -> challenging to be accurately represented, and finer model resolution

Page 3, line 6: low -> lower

Page 3, lines 14-16: Add references.

Page 3, line 16: characterized -> described

Page 8, line 12: transported -> transported over

Page 9, lines 22-23: (e.g., Sinha et al. (2004); Mari et al. (2008)). -> (e.g., Sinha et al., 2004; Mari et al., 2008). Please check the manuscript for similar instances.

Page 10, line 16: our three sites -> the three sites

Page 10, line 16: detected -> the detected

Page 11, line 23: profile -> vertical profile

Please replace all instances of "Brunt-Viäsälä" in the manuscript with "Brunt-Väisälä".

Page 19, Figure 13: dash -> red dash, please also provide information about the black dashes.

Page 22, line 9: If we we assume -> If we assume

Page 22, line 10: impact due to each event STT event -> impact due to each STT event

Page 22: empirically-derived threshholds -> empirically-derived thresholds

Page 22: Comparison with ERA-Interim reanalysis data -> Analysis of the ERA-Interim reanalysis data

Akritidis, D., Pozzer, A., Zanis, P., Tyrlis, E., Škerlak, B., Sprenger, M., and Lelieveld, J.: On the role of tropopause folds in summertime tropospheric ozone over the eastern Mediterranean and the Middle East, Atmos. Chem. Phys., 16, 14025-14039, doi:10.5194/acp-16-14025-2016, 2016.

Elbern, H., Hendricks, J., and Ebel, A.: A climatology of tropopause folds by global analyses, Theor. Appl. Climatol., 59, 181–200, 1998.

James, P., Stohl, A., Forster, C., Eckhardt, S., Seibert, P., and Frank, A.: A 15-year climatology of stratosphere-troposphere exchange with a Lagrangian particle dispersion model: 2. Mean climate and seasonal variability, J. Geophys. Res., 108, 8522, doi:10.1029/2002JD002639, 2003.

Roelofs, G.-J. and Lelieveld, J.: Model study of the influence of cross-tropopause O3 transports on tropospheric O3 levels, Tellus B, 49, 38–55, 1997.

Škerlak, B., Sprenger, M., and Wernli, H.: A global climatology of stratosphere–troposphere exchange using the ERA-Interim data set from 1979 to 2011, Atmos. Chem. Phys., 14, 913–937, doi:10.5194/acp-14-913-2014, 2014.

Škerlak, B., Sprenger, M., Pfahl, S., Tyrlis, E., and Wernli, H.: Tropopause Folds in ERA-Interim: Global, Climatology and Relation to ExtremeWeather Events, J. Geophys. Res.-Atmos., 120, 4860–4877, doi:10.1002/2014JD022787, 2015.

Sprenger, M., Croci Maspoli, M., and Wernli, H.: Tropopause folds and cross-tropopause exchange: A global investigation based upon ECMWF analyses for the time period March 2000 to February 2001, J. Geophys. Res.-Atmos., 108, 8518, doi:10.1029/2002JD002587, 2003.

Terao, Y., Logan, J. A., Douglass, A. R., and Stolarski, R. S.: Contribution of stratospheric ozone to the interannual variability of tropospheric ozone in the northern extratropics, J. Geophys. Res., 113, doi:10.1029/2008jd009854, http://dx.doi.org/10.1029/2008jd009854, 2008.

---

## Referee Comment (RC2) · Anonymous Referee #2 · 6 Feb 2017

**Review for**

**Stratospheric ozone intrusion events and their impacts on tropospheric ozone**

**by Greenslade et al.**
* * *
**Synopsis:**

The authors present an observation-based method to estimate the total stratospheric ozone flux in the Southern Ocean. I think the approach is interesting and complement some model-based methods, and is also of interest to the readership of ACP. However, the method comes with some major uncertainties and I wonder whether an extrapolation to the whole Southern Ocean from only three measurement sites is reasonable. My major concerns are listed below, and based on them I only recommend the manuscript ready for publication in ACP if a carefully revised manuscript is provided.

**Major Concerns:**

**1. Extrapolation to Southern Ocean:** The authors look at three measurement sites (Davis, Macqaurie, and Melbourne) in the Southern Ocean (SO), and then extrapolate their results to the whole SO. I don't think that this is valid. I think there ia quite a lot of spatial and temporal variability that gets neglected in doing so. To make my point more clearly, I copy a figure (Fig. 16) from Skerlak et al. (2014) here:

[Figure]

It shows the seasonally averaged STT ozone flux for the period 1979-2011. Evidently, there is a lot of spatial and temporal variability. The next figure (Fig. 17) from Skerlak et al. (2014) shows the estimated ozone flux into the PBL, which exhibits a still stronger variability. Hence, I think the authors must be rather hesitating in extrapolating their results. I suggest to restrict the conclusions

about the STT flux more to the regions around the three measurement sites. It will still be possible to compare the values, e.g., with the values in Skerlak et al. (2014).

**2. Transport aspect:** An aspect that is not sufficiently discussed in the manuscript is the transport of the ozone-rich air from its crossing to the measurement site. For instance, in Figure 5 the authors show an STT event and the geopotential height field at 500 hPa. A nice cut-off low pressure system is discernible in the geopotential. But it is not clear whether the STT event really occurred below this cut-off. In fact, it could have happened quite a distance away from it and the be advected to this place. I would argue that the transport aspect become more important if an STT event is detected at middle or lower-tropospheric levels, i.e., when it is rather 'detached' from the tropopause above. As an example, the following study shows that the crossing of the tropopause takes place in the western North Atlantic but an ozone signal is discernible in the profile over western Europe:

*Trickl, T. et al. "How stratospheric are deep stratospheric intrusions? LUAMI 2008." Atmospheric Chemistry and Physics 16.14 (2016): 8791-8815.*

I think the authors should more carefully discuss this aspect of STT event. Possibly, the do a short literature review dealing with ozone transport and the long-range character of stratospheric intrusions. It would also be interesting, and relevant to this manuscript, how long signals in stratospheric ozone remain discernible in an atmospheric column after the air parcels have crossed the tropopause.

**3. Uncertainty:** The method comes with quite a few uncertainties! I list some of them:

- P7,L30: "STT events at altitudes below 4 km are removed to avoid surface pollution, and events within 0.5 km of the tropopause are removed to avoid false positives induced by the sharp transition to stratospheric air." → I see the problem with the near-surface STT events. But still, even at this low altitude it could be due to a stratospheric intrusion. Further, I expect quite some ozone flux to be across the tropopause without a very clear peak-like structure in the profile. This could, e.g., be the case if the ozone flux is more related to a continuous 'diffusion' of ozone across the tropopause in contrast to an ozone flux going along with a coherent cross-tropopause air streams in distinct weather systems.

- P7,L9-12: "This estimate is conservative because it does not take into account any ozone enhancements outside of the detected peak that may have been caused by the STT, and also ignores any enhanced ozone background amounts from synoptic-scale stratospheric mixing into the troposphere." → The ozone background is also enhanced in mixing across the troposphere, or the background at any of the stations is enhanced by STT events taking place outside its 'range'.

- In section 5 (P19,L9) the overall ozone flux is determined as the product of the monthly likelihoods of STT (f), the monthly mean fraction of an ozone column attributed to stratospheric ozone (I) and the mean tropospheric ozone column (Omega). All these factors come with a lot of uncertainty! Be it due to the method applied, or the spatial and temporal variability.

- P9,L16: "While ozone production occurs in some biomass burning plumes, this is not always the case; therefore ozone perturbations detected during transported smoke events may or may not be caused by the plume. For this reason all detected STT events found near smoke plumes are flagged." → These events are not included in the calculation of the ozone flux, but still they could be of relevance!

- **P9,L7-9:** " We use the 99th percentile because at this point the filter locates clear events with no obvious false positives. Event detection is highly sensitive to this choice; for example, using the 98.5th percentile instead increased detected events by 10 (22%) at Davis, 19 (40%) at Macquarie Island, and 24 (33%) at Melbourne." → Does this mean that with a 98.5th percentile, some of the events are clear false positives? Wo do you decide that this is the case? I am not sure whether this is obvious. In short, an additional uncertainty of the method.

Given all these uncertainties, the estimate of the total STT flux based on the ozone profiles must be rather conservative and going along with a big overall uncertainty! This is already discussed by the authors, i.e., they are fully aware of it. What I would, however, suggest is a separate section (or extended paragraph) where all uncertainties are presented and, if possible, quantified.

**Minor Comments:**

- **P2,L13-15:** "While models show decreasing tropospheric ozone due to stratospheric ozone depletion propagated to the upper troposphere through vertical mixing (Stevenson et al., 2013), recent work based on the Southern Hemisphere ADditional OZonesonde (SHADOZ) network suggests increasing upper tropospheric ozone near southern Africa, most likely due to stratospheric mixing (Liu et al., 2015; Thompson et al., 2014)" → Simplify sentence structure! Rephrase.

- **P2,L24:** " excedes" → "exceeds"

- **P2,L29:** "STT is responsible" → "STT to be responsible"

- **P3, L4:** "mixing across the tropopause mainly caused by the jet streams" → a little strange formulation. Mixing is not caused by the jet streams; maybe you can write that it is associated by the jet streams.

- **P3,L10-11:** "A big influence on the high surface ozone concentrations over the eastern Mediterranean is stratospheric mixing and anticyclonic subsidence (Zanis et al., 2014)" → "A big influence on the high surface ozone concentrations over the eastern Mediterranean can be attributed stratospheric mixing and anticyclonic subsidence (Zanis et al., 2014)"

- **P3,L11-12:** The authors might want to consider the following studies dealing with STT and ozone fluxes over the eastern Mediterranean:

*Tyrlis, E., B. Škerlak, M. Sprenger, H. Wernli, G. Zittis, and J. Lelieveld (2014), On the linkage between the Asian summer monsoon and tropopause fold activity over the eastern Mediterranean and the Middle East, J. Geophys. Res. Atmos., 119, 3202–3221, doi:10.1002/2013JD021113.*

*Akritidis, D. et al. "On the role of tropopause folds in summertime tropospheric ozone over the eastern Mediterranean and the Middle East." Atmospheric Chemistry and Physics 16.21 (2016): 14025-14039.*

- **P3,L14-15:** "The strength (ozone enhancement above background levels), horizontal scale, vertical depth, and longevity of these intruding ozone tongues vary with weather, topography, and season." → This is a rather general statement. What do you mean with weather?

- **P3,L30-33:** How relevant is it for the reader to know how the ozone mixing ratio is quantified? If not relevant, I would remove this sentence. It sounds rather technical!

**- P4,L8-9:** "Characterisation of STT events requires a clear definition of the tropopause. The two most common tropopause height definitions are the standard lapse rate tropopause (WMO, 1957) and the ozone tropopause (Bethan et al., 1996)." → I would mention already at this place the dynamical tropopause which is defined by means of a potential vorticity iso-surface. I would guess it to be rather similar to the ozone tropopause, but to differ from the WMO one.

**- P4,L17:** "The ozone tropopause can be less robust during stratosphere-troposphere exchange;" → What does 'robust' mean? What defines whether a tropopause is robust or not? The ozone tropopause certainly allows for much more details (and a much more complicated structure) than the WMO tropopause. But I would not say that it is less robust because of this!

**- P4,L19-21:** "In this work, the lower of these two tropopause altitudes is used. This choice avoids occasional unrealistically high tropopause heights due to perturbed ozone or temperature measurements in the ozonesonde data." → I feel a little uncomfortable by this definition! The two definitions of the tropopause are rather different, and by simply taking the lower one seems 'dangerous'. The authors should motivate this approach more clearly. At least, I would like to know how often the ozone tropopause 'wins' and how often the WMO one. I would expect the ozone tropopause most often to be at lower heights than the WMO one! Correct?

**- P7,L22-23:** "The interpolated profiles are then bandpass-filtered using a Fourier transform to retain perturbations with vertical scales between 0.5 km and 5 km (removing low and high frequency perturbations)" → I see the 0.5-km threshold. What exactly is the aim of the low-pass filtering threshold (5 km). A more clear description would be helpful.

**- P7,33-34:** "The STT event is confirmed if the perturbation profile drops below zero between the ozone peak and the tropopause" → Why does have to drop below zero?

**- P8,Figure 3:** Just for curiosity: In the ozone profile the Ozone mixing ratio (OMR) is rather low right above the identified STT event. The OMR is higher than immediately below the STT event. Is their a simple reason why the OMR is so low right above the STT peak?

**- P9,L16:** "all detected STT events found near smoke plumes are flagged." → What does 'near' mean?

**- P 10,L15-16:** "Data from the European Centre for Medium-range Weather Forecasts (ECMWF) Interim Reanalysis (ERA-I) (Dee et al., 2011) product are used for synoptic-scale examination of weather patterns over our three sites on dates matching detected STT events" → Please rephrase! For instance: "Synoptic-scale weather patterns are examined based on the ERA-Interim dataset (Dee et al., 2011). More specifically, the ERA-I products over the three sites are used on dates matching detected STT events.

**- P11,L17+26:** Here, the STT event is subjectively linked to a meteorological feature, a cut-off low-pressure system. The argument is not very 'strong'. I don't think that a lowering of the tropopause itself can explain the flux of stratospheric ozone. It would be interesting to see a vertical cross section the cut-off low, with tropopause height included. Is the cut-off low eroded away from below, or how does the flux across the tropopause in the cut-off low really takes place? Some further thoughts on this might be helpful. The following paper might be a starting point:

*Stohl, A., et al. "Stratosphere-troposphere exchange: A review, and what we have learned from STACCATO." Journal of Geophysical Research: Atmospheres 108.D12 (2003).*

**- P12, Figure 6:** In this Figure 6 relative humidity is shown in addition to ozone. I wonder whether this signal is available for all soundings, if if so, whether it would be worthwhile to include it into the identification method of STT events. Of course, this would be based on the dryness of stratospheric air.

**- P12,L10:** "This summertime peak is due to a prevalence of summer low-pressure storms and fronts" → Please add a reference which supports this statement. Reutter et al. (2015) doesn't deal with that, as far as I see. Further, I don't see why the summer low-pressure storms and fronts increase turbulence. Here too, a reference might be appropriate.

**- P12,L11-P13,L1:** "At Davis, there is increased Antarctic winter activity, which may be due to the polar vortex and it's associated lowered tropopause and increased turbulence" → Again, I think that this statement must be supported by a reference. Why is Antarctic winter activity increased (by the way, what is meant with 'activity'?)? Why is this related to the polar vortex? And, why is there enhanced turbulence in this case? As a starting point, the authors might want to look at the SH climatologies of extratropical cyclones:

*Jones, David A., and Ian Simmonds. "A climatology of Southern Hemisphere extratropical cyclones." Climate Dynamics 9.3 (1993): 131-145.*

*Simmonds, Ian, and Kevin Keay. "Mean Southern Hemisphere extratropical cyclone behavior in the 40-year NCEP–NCAR reanalysis." Journal of Climate 13.5 (2000): 873-885.*

**- P13,L3-5:** "This seasonality is not seen in the recent ERA-Interim tropopause fold analysis performed by Škerlak et al. (2015), where a winter maximum over Australia can be seen (although in the subtropics only - from around 20 S to 40 S)" → This comparison is somewhat misleading, because Davis is located at 69 S, but the authors mention that the fold maximum in Skerlak et al. (2015) occur between 20-40 S. I wonder whether the authors would benefit from the following data source that provides global monthly climatologies of several features (among them tropopause folds):

*Sprenger, M. et al. "Global climatologies of Eulerian and Lagrangian flow features based on ERA-Interim reanalyses." Bulletin of the American Meteorological Society 2017 (2017).*

**- P13, Figure 7:** At Davis no STT events are associated with fronts, quite in contrast to the two neighboring months. I wonder whether this is correct!

**- P14,L10-11:** "and assume that the vertical temperature gradients within the intrusion respond most rapidly to transported heat, which is an additional characteristic of stratospheric air" → I do not understand this statement! Which transported heat? Why do the vertical temperature gradients respond most rapidly to it? Please explain!

**-P15,L24-28:** "There is no clear relationship between meteorological conditions and event altitude." → Could this be related to the fact that the transport (and time) between the tropopause crossing and the STT event in the ozone profile is neglected? The same question can be asked whether for the link to meteorology at the the other two sites.

**- P15,L29:** "Simulation of southern mid-latitude ozone columns" → While reading this section I wondered what it has to do with the topic of the paper, i.e., with STT events. Of course, later (in section 5) it becomes perfectly clear. But, maybe its better to add one or two sentences that make it clear to the reader from the beginning. One or two introductory sentences to section 4 might be OK.

**- P16,L12:** "the r2 values decrease to .07, .11, .30 respectively" → As far as I can see these de-seasonlized values are particularly relevant. Right? But the explained variance (equal to $r^2$) seems to be rather low?

**- P17,L1-12:** At first reading I wondered whether I should worry about GEOS-Chem's inability to capture STT events. I also wondered whether the underestimation of ozone in the lower troposphere ( up to 6 km) has something to do with the model's deficiency in capturing the STT events. Only in the next section 5 I learned what the GEOS-Chem's ozone columns are used for. I guess that the reading of section would have been more 'rewarding' if I had knew in advance why we are now looking at these GEOS-Chem simulations. In short, as before I would suggest to guide the reader a little more clearly and help him/her not to be mislead... Some short introductory sentences to section 4 might be sufficient to do so.

**- P20,L1-7:** Here some aspects of limitations of the method are discussed, or the rather strong assumptions are discussed and what could be done in a next step. As suggested in my major concerns, I would appreciate if all these discussions are concisely, but in sufficient depth, discussed at one place in the manuscript.

**- P21, Figure 16:** The ozone flux in the lower panel essentially has the same shape as the f part in the upper panel. Is it, hence, correct to say that the estimated ozone flux is essentially determined by this factor (the monthly frequency of STT events)? This might be relevant if a more detailed analysis of the uncertainties of the method is provided.

**- P21,L7-13:** Here the estimates from the three sites are extrapolated to the Southern Ocean. As stated in my major concern, this most likely is too 'strong'. I would prefer if the fluxes at the three sites are taken to estimate the STT ozone flux in the regions around them, and then compared to the fluxes from other studies (Stevenson et al., 2006; Sprenger et al., 2003). In addition to the studies listed, I would also look at the following study:

*Skerlak, B., M. Sprenger, and H. Wernli. "A global climatology of stratosphere-troposphere exchange using the ERA-Interim data set from 1979 to 2011." Atmospheric Chemistry and Physics 14.2 (2014): 913.*

Possibly, from these global climatologies some regional estimates of ozone flux can be extracted and compared to the fluxes at the three sites.

**- P22,L5-6:** "Comparison with ERA-Interim reanalysis data suggested the majority of events were caused by turbulent weather in the upper troposphere due to low pressure fronts, followed by cut-off low pressure systems" → The term 'turbulent weather' is not very meaningful! I think the link to turbulence could simply removed from the manuscript. It is not really shown in the paper, and ERA-Interim might not be the best dataset to address it due to its coarse resolution. I think just listing the weather systems would be OK, although the link to them could be more deeply discussed in the paper.

---

## Referee Comment (RC3) · Anonymous Referee #3 · 23 Feb 2017

The paper by Greenslade and coauthors presents an analysis of ozone soundings from three locations between -31°S and -69°S over several years. The authors analyse the profiles for ozone enhancements in the troposphere, which they link to stratosphere-to-troposhpere exchange associated with cut-off lows and cyclones. Based on these enhancements they estimate the fraction of excess ozone from the stratosphere as percent of the tropospheric ozone column. They calculate a flux of ozone for the southern hemisphere by using tropospheric columns from GEOS-Chem times this fraction times the frequency of occurrence of STT events. This flux estimate is more than an order of magnitude lower than estimates from literature using various techniques. The authors conclude that these differences are due to conservative estimates of several thresholds and assumption regarding their method.

Observationally based estimates of ozone transport especially in the southern hemi-

sphere are sparse and therefore valuable. However, the authors provide a flux estimate which is far from other studies, probably due to the sparse spatial and temporal coverage. If this however is the case, the method is simply not applicable in this case and the deduced flux does not mean anything. If the method is valid for the given data set I miss a careful analysis of the reasons for the discrepancy. Therefore I don't see the paper as an ACP paper in the current form.

Major comments:

1) Overall the manuscript leaves me a bit puzzled, since I'm not sure what to take out of this work. The authors state that their value of the ozone enhancement fraction might be largely underestimated by a factor of ten. If this is the case it is difficult to get the benefit of the study. Though the approach is reasonable, maybe the statistics and spatial coverage is to small to cover the full variability and frequency of occurrence of STT events for a quantitative flux calculation for the southern hemisphere. If the difference between observations and models is really between factors 30-200 depending on the reference (p.21, l.8-12), this needs more clarification than simply replacing observation with model results to find agreement with other studies. I found this approach at least very questionable.

2) Further: I missed a quantification of the uncertainties. This is partly done in section 2.5, but it is e.g. not clear why a threshold of 99% is the best choice nor which factor specifically leads to the very low flux estimate. I suggest the authors use a bunch of northern hemispheric sondes with higher spatial and temporal density to gauge their approach before applying it to the southern hemisphere.

Minor points: p.1,l.5: Please add the period of observations

p.4, l. 9: At least mention the dynamical tropopause, it is more common than ozone...

p.4, l.11: Correct definition of the thermal tropopause "... provided the lapse rate averaged between this altitude ..."

[Figure]

p.4, l.20 (also Fig.1): The tropopause definitions are mixed here. Why do the authors not include the dynamical definition? The effect of the pure lapse rate criterion is misleading under specific synoptic conditions as correctly stated. This might explain the very low cases in Fig.1.

p.6, l.15: How many model levels are between the sea level and 14 km? How many model levels are between 8 and 14 km and how are sonde and profile data compared? Pointwise or vertically averaged to fit the model levels?

p.6, l.15+: The sonde profiles are compared against model data of 2 x 2.5 degrees grid sizes (and the vertical model resolution). How well does the model resolve the soundings? How do the authors estimate the fraction of ozone transport which is missed due to unresolved structures? Why do the authors don't interpolate to the time window of the sounding (or at least use the according model time step)?

p.11, l.17: Even if you did a subjective method, could you explain a bit more in detail in the manuscript, how you distinguished different potential situations? What are upper tropospheric "low pressure fronts"? Tropospheric intrusions (3D!) in the stratosphere or stratospheric cut-offs (fully detached)?

p.11,l.20: What are "ozone folds" without other sources of upper tropospheric turbulence and how are these related to the polar vortex?

p.11, l.25: Explain: "...ozone enhancements derived from dry stratospheric air..." - didn't you use the methods and criteria from sec.2?

p.12, Fig.6. and related discussion (shortly before section 3): Please show a cross section of PV since most likely the ozone peak is related to a tropopause fold.

p.12, last line: What is meant with increased winter activity? More tropopause folds, stronger tropospheric winds, cyclone activity, etc...? Please be more precise. How do you expect the vortex to affect the tropopause?

p.14, l.6-20: Why do you use N2 as indicator? The relation you found is interesting,

but not necessarily valid since stability is not conserved. Why should it be 'retained' when crossing the thermal tropopause? In general the thermal tropopause is ill defined under these conditions. Why not simply taking PV for this excercise or humidity as a measurement based quantity?

Fig.11 and related discussion: Couldn't you provide scatter plots (or Taylor diagram) of the column ozone between sondes and model?

Fig.3 caption: Units: concentration or mixing ratio?

---

## Author Comment (AC1) · 26 May 2017

We thank the three reviewers for their detailed, thoughtful and insightful reviews which have improved the quality of our manuscript. In particular, we value their critique on our technique to determine a whole hemisphere flux, which has lead to an adjustment of our thresholds to incorporate more data and improve robustness in this data sparse Southern Hemisphere region. We write our detailed responses to each comment in blue text (and quoted text in magenta), directly below the specific reviewer comment.

Attachments include the updated manuscript, the author response document, the difference document generated by latexdiff, and a supplementary file.

Regards, Jesse Greenslade on behalf of all the coauthors.

[Figure]

Please also note the supplement to this comment:
http://www.atmos-chem-phys-discuss.net/acp-2016-1124/acp-2016-1124-AC1-supplement.pdf

[revised manuscript text omitted]

**Author's Response**

We thank the three reviewers for their detailed, thoughtful and insightful reviews which have improved the quality of our manuscript. In particular, we value their critique on our technique to determine a whole hemisphere flux, which has lead to an adjustment of our thresholds to incorporate more data and improve robustness in this data sparse Southern Hemisphere region. We write our detailed responses to each comment in blue text (and quoted text in magenta), directly below the specific reviewer comment.

**Anonymous Referee 1**

Referee Notes:

The current study presents a method to identify stratosphere-to-troposphere transport (STT) events and estimate the associated ozone flux to the troposphere, based on ozonesonde profiles from three sites located in the Southern Hemisphere extratropics. Subsequently, the seasonality of STT events is determined, as well as the favorable synoptic conditions. Based on the stratospheric contribution to tropospheric ozone column estimated from the ozonesondes, the GEOS-Chem simulated tropospheric ozone columns are extrapolated to assess the stratospheric contribution over the Southern Ocean region. As the STT is of great importance for the tropospheric ozone budget and variability, and the number of relevant studies (both observational and modeling) for the examined region is limited, I find the topic of the paper within the scope of ACP. On the other side, there are several issues that need to be addressed before consideration for publication in ACP.

Major Comments:

1) Calculating the 99th percentile from the perturbation profiles over that layer (2 to 1 Km below the tropopause) is a fairly strict criterion. Wouldn't this threshold choice avoid the selection of deeper stratospheric intrusion events as "STT events"? This should read as 2 km above the surface to 1 km below the tropopause since, as presently written, it implies a one kilometre range which would miss deeper intrusions. The text has been updated as follows on page 8, line 7: "... (2 km above the earth's surface to 1 km below the tropopause)." and on page 24, line 3: " ... profiles between 2 km above the earth's surface and 1 km below the tropopause.".
Have you consider modifying this criterion, and include others (e.g. significant negative O3 relative humidity correlation values above a threshold) to minimize false STT detection?
We investigated modifying this 99$^{th}$ percentile threshold in detail and found that lowering the limit to the 95$^{th}$ percentile is the optimal balance between including as many detections as possible while minimising false positives. Thus, we use the 95$^{th}$ percentile limit in our revised manuscript. Following an inspection of the parsed data, we found that lowering the threshold further resulted in too many clearly incorrect "O3 events" being incorporated into the results. We prefer to only include events which are clear STE events, which we accept could result in an underestimate of STE flux, than including data which are clearly spurious. Regarding use of humidity, this parameter is known to be biased low in the upper troposphere. To some extent this can be corrected for with the type of radiosonde (RS-90) which we used. However we have not performed this correction because it is difficult to apply uniformly across all sites due to seasonal variation in bias that is driven by the seasonal variation in tropopause temperature.

[Figure]

**Fig. 3.** Our responses to reviewers comments and suggestions

**Supplementary: Stratospheric ozone intrusion events and their impacts on tropospheric ozone in the Southern Hemisphere**

Jesse W. Greenslade[1], Simon P. Alexander[2,3], Robyn Schofield[4,5], Jenny A. Fisher[1,6], and Andrew K. Klekociuk[2,3]

[1]Centre for Atmospheric Chemistry, School of Chemistry, University of Wollongong
[2]Australian Antarctic Division, Hobart
[3]Antarctic Climate and Ecosystems Co-operative Research Centre, Hobart, Australia
[4]School of Earth Sciences, University of Melbourne
[5]ARC Centre of Excellence for Climate System Science, University of New South Wales
[6]School of Earth & Environmental Sciences, University of Wollongong

*Correspondence to:* Jesse Greenslade (jwg366@uowmail.edu.au)

**Fig. 4.** Supplementary document

**Supplement:**

**Supplementary: Stratospheric ozone intrusion events and their impacts on tropospheric ozone in the Southern Hemisphere**

Jesse W. Greenslade[1], Simon P. Alexander[2,3], Robyn Schofield[4,5], Jenny A. Fisher[1,6], and Andrew K. Klekociuk[2,3]

[1]Centre for Atmospheric Chemistry, School of Chemistry, University of Wollongong
[2]Australian Antarctic Division, Hobart
[3]Antarctic Climate and Ecosystems Co-operative Research Centre, Hobart, Australia
[4]School of Earth Sciences, University of Melbourne
[5]ARC Centre of Excellence for Climate System Science, University of New South Wales
[6]School of Earth & Environmental Sciences, University of Wollongong

*Correspondence to:* Jesse Greenslade (jwg366@uowmail.edu.au)

[Figure]

**Figure S1.** Example detection of biomass burning influence using AIRS total column CO. The top panel (17 October 2007) shows a day when ozone above Melbourne (purple dot) could have been caused by a transported biomass burning plume, and so was flagged in subsequent analysis. The bottom panel (3 February 2006) shows a day when Melbourne ozone was not influenced by transported smoke.

**S1    Weather and Smoke analysis**

Figure S1 contrasts two days; one with and one without signs of biomass burning influence near the Melbourne site (purple circle). On 17 October 2007 (top) elevated CO suggests the site may have been influenced by long-range transport from African and/or South American biomass burning. In contrast, on 3 February 2006 (bottom) CO columns across the SH show no influence from biomass burning.

Figure S2 (left) shows the vertical ozone profile on 3 February 2005. The tropopause was between 400 and 500 hPa and ozone in the upper troposphere was anticorrelated with relative humidity, suggesting the ozone enhancements are due to dry stratospheric air. An ozone intrusion into the troposphere at ~520 hPa was identified by our detection algorithm. The right panel shows the concurrent synoptic weather system, a cut-off low pressure system that caused a large storm and lowered the local tropopause height for several days. The flux of stratospheric ozone into the troposphere associated with this event,

[Figure]

**Figure S2.** (Left) Vertical profile of ozone (black), relative humidity (blue), and temperature (red) measured by ozonesonde over Melbourne on 3 February 2005. The detected ozone STT event is highlighted in pink. Tropopause heights using both the ozone definition (black dashed line) and lapse rate definition (red dashed line) are also shown. (Right) Geopotential heights at 500 hPa from the ERA-Interim reanalysis, with wind vectors over-plotted. Also shown is the 1 PVU contour line (purple).

calculated using the method shown in Sect. 2.3 of the parent document, was at least $3.1 \times 10^{11}$ molecules cm$^{-3}$, or 8% of the tropospheric ozone column.

Figure S3 (left) shows the ozone profile over Melbourne on 13 January 2010. The tropopause was higher on this date (120-160 hPa). Using our algorithm, we detected an ozone intrusion centred around 200 hPa. As before, ozone anti-correlation with relative humidity provides further evidence that the elevated ozone was stratospheric in origin. In this profile, there was clear separation between the detected intrusion (highlighted in pink) and the ozone tropopause (black dashed line), which indicates that the sonde passed through regular tropospheric air after hitting a stratospheric intrusion but before reaching the tropopause. The right panel shows that this event was associated with a trough (front) of low pressure passing over south eastern Australia. This front travelled from west to east and caused a wave of lowered tropopause height. Frontal passage is a known cause of STT as stratospheric air descends and streamers of ozone-rich air break off and mix into the troposphere (Sprenger et al., 2003).

[Figure]

**Figure S3.** Same as Fig. S2 but for 13 January 2010. Also shown in this figure is the 2 PVU contour (white), often used to determine dynamical tropopause height.

**S2 Southern Ocean extrapolation**

**S2.1 Outline**

We use simulated tropospheric ozone columns from GEOS-Chem to extrapolate the ozonesonde-based estimates over a large area of the Southern Ocean encompassing our three measurement sites. Figure S4 shows the region defined by latitudes 79° - 28° S, and longitudes 53° - 175° E.

Figure S5 (upper panel) shows the factors I, P, and $\Omega_{O_3}$ which are used along with the assumed event lifetime to estimate the STT flux. The tropospheric ozone and area of our region is calculated using the output and surface area from GEOS-Chem over the Southern Ocean grid boxes along with the molecules cm$^{-2}$ per month calculations, along with ozone molar mass of 48 g mol$^{-1}$.

It is worth noting that this extrapolation is very simplistic and is performed as an example of how the seasonal ozone STT calculations could be used. A more spatially resolved estimate could be determined by dividing the Southern Ocean region into longitudinal and latitudinal bins for calculating the average $\Omega_{O_3}$ from GEOS-Chem, as well as applying latitudinal gradients to $P$ and $I$ based on their values at the three sonde release sites, and adding longitudinal variability based on seasonal stratospheric wind jet streams (Baray et al., 2012; Škerlak et al., 2015). An improved estimate of event lifetime and parameterisation of how many events may be occuring simultaneously could also be addressed, however this is beyond the scope of this work and in any case would not address all the limitations of the estimate provided below.

[Figure]

**Figure S4.** Region used for large SO estimation of STT flux

**S2.2 Results**

Fig. S5 (lower panel) shows the results of the calculation when we choose two days for our flux estimation, with the range
10   shown in representing the values calculated if we assume events last one day (upper bound of estimated flux) or one week
(lower bound of estimated flux). Previous studies have found STT ozone fluxes in the SH extratropics are largest from autumn
or winter to early spring (Olsen, 2003; Škerlak et al., 2015; Liu et al., 2016). Although these are based on dominating STT
systems further north than the area we examine here, see the main text for more details. During the SH winter, we find
the highest tropospheric $\Omega_{O3}$ but a relatively low STT flux due to reduced event frequency. Our results suggest instead that
15   the ozone flux associated with STT events (at least those due to tropopause folds) is largest in austral summer (December-
March), primarily due to an increased frequency of STT detections during these months. It is possible that our estimated event
frequencies are too low in late winter-early spring as some legitimate STT events may have been excluded due to coincident
smoke plumes.

Summing the monthly estimated fluxes shown in Fig. S5 over the year, we find from this estimate that STT events may be
20   responsible for $\sim 7.5 \times 10^{16}$ molecules cm$^{-2}$ yr$^{-1}$ of the tropospheric ozone over the Southern Ocean, equivalent to 75 Tg
yr$^{-1}$. 2-16 $\times 10^{16}$ molecules cm$^{-2}$ of stratospheric based ozone is estimated over the southern ocean throughout the year. Our
estimate is hard to directly compare to prior work that suggests global gross STT fluxes of 550 Tg yr$^{-1}$ (Stevenson et al.,
2006) and net downward STT fluxes of 75 Tg yr$^{-1}$ (Sprenger et al., 2003). This is due to the high uncertainties involved in
calculation as well as the specific regions which have few other measurements available.

[Figure]

**Figure S5.** (Top) The three quantities used to calculate the total Southern Ocean ozone flux from STT events. The tropospheric ozone column $\Omega_{O_3}$ (black, left axis) is from GEOS-Chem, while the STT probability $P$ (magenta, right axis) and impact $I$ (teal, right axis) are from the ozonesonde measurements. The STT impact is multiplied by 10 to better show the seasonality. (Bottom) Estimated contribution of STT to tropospheric ozone columns over the Southern Ocean. The shaded area shows the uncertainty as calculated in Sect. 6 of the parent document, with the -dashed lines showing the range of values when assuming events last one day (upper dashed line) up to one week (lower dashed line).

---

## Author Comment (AC2) · 7 Jun 2017

**Author's Response**

We thank the three reviewers for their detailed, thoughtful and insightful reviews which have improved the quality of our manuscript. In particular, we value their critique on our technique to determine a whole hemisphere flux, which has lead to an adjustment of our thresholds to incorporate more data and improve robustness in this data sparse Southern Hemisphere region. We write our detailed responses to each comment in blue text (and quoted text in magenta), directly below the specific reviewer comment.

**Anonymous Referee 1**

**Referee Notes:**

The current study presents a method to identify stratosphere-to-troposphere transport (STT) events and estimate the associated ozone flux to the troposphere, based on ozonesonde profiles from three sites located in the Southern Hemisphere extratropics. Subsequently, the seasonality of STT events is determined, as well as the favorable synoptic conditions. Based on the stratospheric contribution to tropospheric ozone column estimated from the ozonesondes, the GEOS-Chem simulated tropospheric ozone columns are extrapolated to assess the stratospheric contribution over the Southern Ocean region. As the STT is of great importance for the tropospheric ozone budget and variability, and the number of relevant studies (both observational and modeling) for the examined region is limited, I find the topic of the paper within the scope of ACP. On the other side, there are several issues that need to be addressed before consideration for publication in ACP.

**Major Comments:**

1) Calculating the 99th percentile from the perturbation profiles over that layer (2 to 1 Km below the tropopause) is a fairly strict criterion. Wouldn't this threshold choice avoid the selection of deeper stratospheric intrusion events as "STT events"?
This should read as 2 km above the surface to 1 km below the tropopause since, as presently written, it implies a one kilometre range which would miss deeper intrusions. The text has been updated as follows on page 8, line 7:  "... (2 km above the earth's surface to 1 km below the tropopause)." and on page 24, line 3: " … profiles between 2 km above the earth's surface and 1 km below the tropopause.".
Have you consider modifying this criterion, and include others (e.g. significant negative O3 relative humidity correlation values above a threshold) to minimize false STT detection?
We investigated modifying this $99^{th}$ percentile threshold in detail and found that lowering the limit to the $95^{th}$ percentile is the optimal balance between including as many detections as possible while minimising false positives. Thus, we use the $95^{th}$ percentile limit in our revised manuscript. Following an inspection of the parsed data, we found that lowering the threshold further resulted in too many clearly incorrect "O3 events" being incorporated into the results. We prefer to only include events which are clear STE events, which we accept could result in an underestimate of STE flux, than including data which are clearly spurious. Regarding use of humidity, this parameter is known to be biased low in the upper troposphere. To some extent this can be corrected for with the type of radiosonde (RS-90) which we used. However we have not performed this correction because it is difficult to apply uniformly across all sites due to seasonal variation in bias that is driven by the seasonal variation in tropopause temperature.

2) The seasonality of STT events presented in Fig. 7 is not in line with the findings of Škerlak et al. (2015) for the examined region. How are your results (STT seasonality) compared with other modeling studies (Elbern et al., 1998; Sprenger et al., 2003)?

We have added discussions between the seasonalities reported at Melbourne, Macquarie Island and Davis in this manuscript and results from Wauben et al., 1998, Sprenger et al., 2003, and Škerlak et al., 2014 and 2015. The measurement sites are not in the regions which have a clear winter maximum seen in figure 1 of Sprenger et al., 2003, and the large scale winter maximum shown by all three studies seems to be dominated by the system in that region – this has now been noted in the text as follows on page 11, line 16: "...The SH summer maximum we see for STT ozone flux can also be seen in Fig. 16 of Škerlak et al. (2014), which shows seasonal flux over the southern ocean, although this is less clear over Melbourne. This seasonality is not clear in the recent ERA-Interim tropopause fold analysis performed by Škerlak et al. (2015), where a winter maximum of ozone fold frequency (~0.5% more folds in winter) over Australia can be seen to the north of Melbourne. Their work seems to show slightly higher fold frequencies over Melbourne in summer (Škerlak et al., 2015, Fig. 5), which agrees with our ozonesonde measurements. Their winter maximum is in the subtropics only - from around 20S to 40S, which can be seen as the prevalent feature over Australia in their Fig. 5. Wauben et al. (1998) look at modelled (CTM driven by ECMWF output) and measured ozone distributions and find more SH ozone in the lower troposphere during Austral winter, however they note that the ECMWF fields are uncertain here again due to lack of measurements. Their work shows a generally cleaner lower troposphere in the SH summer but this can not be construed to suggest more or less STT folds in either season. Sprenger et al. (2003) examine modelled STT folds using ECMWF output over March 2000 - April 2001, and show that for this year there is a clear Austral winter maximum, again over the 20S to 40S band. The winter maximum does not include Melbourne, or the southern ocean, which explains why we see a seasonality which disagrees with these prior studies."

Is there any evidence from other studies that STT frequency over the examined region exhibits a maximum during the austral summer (DJF) and not during the austral winter (JJA) when the jet stream is strongest over the broader region?

Fig. 16 in Škerlak et al., 2014 shows the seasonal STT ozone flux, and a summer maximum is apparent over the Southern Ocean. Fig. 5 of Škerlak et al., 2015 may also agree with our ozonesondes, as there appears to be slightly higher summer fold frequencies over Melbourne. Please refer to our response immediately above for the additions to the text which also address this point.

Have you tried to detect STT events from the model results? I guess this is strongly depended to the vertical resolution of the model, but it would be very interesting to see how the observed and modeled STT seasonalities are compared.

Not only the vertical resolution but also the low horizontal resolution of GEOS-Chem output means an analysis would only perceive very large scale STT events, as they would need to span a large portion of 2 degrees latitude by 2.5 degrees longitude. Therefore we are unable to use GEOS-Chem to detect the STT events which our ozonesondes are capable of detecting.

3) To my understanding, using the seasonality of STT events from the three sites to extrapolate model results over the Southern Ocean region is a quite simplified and coarse approach, especially when considering the previous comment.

We agree with this comment. After considering the reviews, we have removed the SO extrapolation and replaced it with a smaller scale examination of STT events in regions surrounding the three release sites. Please refer to the new Figure 1 for the location of these regions.

STT ozone flux near each site has been calculated and compared against Škerlak et al. 2014 on page 20, line 25: "Škerlak et al. (2014) show an estimate of roughly 40 to 150 kg $km^{-2}$ $month^{-1}$ in these regions, over all seasons, (see Fig. 16, 17 in their publication) while we estimate from 0 to 180 kg $km^{-2}$ $month^{-1}$ STT impact, following a seasonal cycle with the maximum in austral summer. We

estimate higher maximum flux over Melbourne, (178, and 150 kg km$^{-2}$ month$^{-1}$ in January and February) than in either Davis (89 kg km$^{-2}$ month$^{-1}$ in March) or Macquarie Island (68 kg km$^{-2}$ month$^{-1}$ in January). Our calculated seasonal contributions, along with total uncertainty are shown in Table 3."

4) Overall, the presentation of the results can be further improved (please check my suggestions further below), as well as the writing of the manuscript.
We thank the reviewer for the comments below which have helped us improved presentation of the manuscript.

**Comments:**

Škerlak et al. (2014) presented an STE climatology using the ERA-Interim data. This study is important not only for the introduction, as it describes the STT climatology for the SH, but for intercomparison of the results also. Similar climatologies can be found in the modeling studies of Roelofs and Lelieveld (1997) and James et al. (2003). Recently, Akritidis et al. (2016) explored the impact of stratospheric intrusions on tropospheric ozone and the associated stratospheric contribution over the eastern Mediterranean and the Middle East region, a task that is relevant with some of the purposes of this study.
We have now referenced Akritidis, and included specific mention of Roelofs and Lelieveld as follows: "… other studies which have found STT ozone fluxes in the SH extra-tropics are largest from autumn or winter to early spring. Roelofs and Lelieveld (1997) use a model with a stratospheric ozone tracer to estimate STT impacts, they see higher SH tropospheric ozone concentrations, as well as STT flux, in the SH winter."
Full comparison of our results against Škerlak et al. (2014, 2015) has been added where possible, including a conversion of our output to Kg per month within each season and the uncertainties as shown in Table 3, and in the added text quoted in response to Major comment 3.
Page 4, lines 3-4: Since the study is based on the ozonesondes launched from the three sites, it is important to present the location of the sites.
A brief description of the sites has been added to the revision: "... Melbourne, a major city in the south east of Australia. Macquarie Island is isolated from the Australian mainland, situated in the remote Southern Ocean and unlikely to be affected by any local pollution events. Davis is on the coast of Antarctica and also unlikely to experience the effects of anthropogenic pollution."

Page 4, line 22: "Figure 1 shows the monthly mean tropopause altitudes at ..", while in Fig. 1 caption is stated "Multi-year monthly median tropopause altitude ..". Is it the mean or the median? Please modify accordingly.
It is the median, Fig. 1 caption has been updated (and please note this is now Figure 2 in the revised manuscript)

Page 5, Figure 1: a) The shadings used to describe the 10th and 90th percentiles are rather confusing. I suggest you replace the shadings with dashed lines (same color as the solid lines).
We have made this change (now Figure 2 in the revision)
b) Increase the range of the vertical axis to show the 10th percentile value for February.
 [please refer to our reply directly below]
c) Is it the case that tropopause drops below 4 km (10th percentile) over Davis?
We looked into this and found a problem with the lapse rate tropopause picking up boundary layer temperature inversions – frequently enough that it showed up in the Davis 10$^{th}$ percentile.
We set a lower limit in our algorithm requiring the lapse rate tropopause (LR TP) to be a minimum

of 4km altitude (rather than the 2km minimum previously), since every LR TP detected below that height was a false positive.
This is mentioned in the text at page 5 line 17: "... We require lapse rate tropopauses to be at a minimum of 4 km altitude"
We are now using the ozone defined tropopause in our event detection algorithm, which means the LR TP no longer affects event detection. The ozone tropopause climatology is now displayed in the revised Figure 2.
What is the minimum tropopause height value over Davis during February?
The lowest TP occurred at Macquarie: at 4.4 km, while Davis got as low as 6.14 km and Melbourne's lowest was 5.81 km.

Page 5, lines 5-6: "This seasonality at the high latitude sites is driven by a decrease in photochemical destruction under the reduced radiation conditions around polar night."
Please include a reference or information about the NOx levels at these sites (if available) to justify this statement.
The seasonality shown in Figure 3 is consistent with remote free tropospheric photochemistry determined by solar radiation availability and temperature, resulting in higher ozone in winter [*Lelieveld and Dentener*, 2000]. NO2 stratospheric observations have been conducted in the Southern hemisphere at Lauder, Macquarie island and Arrival Heights i.e. [*Struthers et al.*, 2004] which displays a winter minima in seasonality consistent with an ozone maxima. We add these references to the text.
Page 6, line 14: It is important to know the vertical resolution of the GEOS-Chem model near the tropopause (although it can partially be seen from Fig. 13), as it is important for the tropopause height detection and the tropospheric ozone column calculations from the model results.
GEOS-Chem has roughly 500m resolution near 10km altitude. We have now noted this in the text in the Model description section at page 7, line 14: "The vertical resolution is finer near the surface at ~60 m between levels, increasing to ~500 m near 10 km altitude."

Page 7, lines 22-23: "The interpolated profiles … high frequency perturbations)." This is a rather brief description of the procedure. A more detailed description including a reference (if available) for the FT application would be necessary.
We have added a reference to Press et al., 1992, along with the following improved description on page 8 line 25: "…*to 14 km altitude. Small vertical-scale fluctuations in ozone, which are captured by the high-resolution ozonesondes, can be regarded as sinusoidal waves superimposed on the large vertical scale background tropospheric ozone. As such, the interpolated profiles are bandpass-filtered using a fast Fourier transform (Press et al., 1992) to retain these small vertical scales, between 0.5 km and 5km…*"

Page 7, lines 27-28: "We next use all the perturbation profiles at each site to calculate the 99th percentile perturbation value for the site". How exactly is this cut-off threshold calculated?
(Please note that in our revision, we now use the 95th percentile as our threshold cut-off as we found this the optimum balance between maximise data acceptance and minimising spurious false positives). Once the perturbation profiles are all created, the filtered interpolated values between 2km and TP-1km are used as the basis for this percentile. This has been added to the text at page 8 line 4: "... For an event to qualify as STT, a clear increase above the background ozone level is needed, as a bandpass filter leaves us with enhancements minus any noise or seasonal scale vertical profile effects. The 0.5 km scale limit is set in order to remove any spikes of ozone which could be considered noise. We next use all the perturbation profiles at each site to calculate the 95th percentile perturbation value for the site. The threshold is calculated from all the interpolated filtered values between 2 km above the surface and 1 km below the tropopause. This is our threshold for tropospheric ozone perturbations, and any profiles with perturbations exceeding this

value in individual ozonesondes are classified as STT events."

In Section 2.5, Page 9, the authors state that is calculated "between 2 km and 1 km below the tropopause". This information should be provided earlier in the manuscript, at the point that the 99th percentile threshold is initially mentioned (Section 2.3).

As per our response to Major Point 1, we have rewritten this sentence in the revision because it was not clearly detailing our analysis procedure.

Page 8, Figure 3: Why the two panels have different units? Are the ozone units of the left panel "1e+12 molecules cm-3"? Please change accordingly the Figure and the Figure caption. mixing ratio -> number density

We have redrawn the plots (now Figure 4) with consistent units.

Page 9, lines 1-2: "For this reason all detected STT events found near smoke plumes are flagged". How is "near" defined?

In this case near is defined as subjectively within 150km, This is now more clear in the text at page 10, line 32: "For this reason all detected STT events found near (within ~150 km of) smoke plumes are flagged, following visual inspection. Removal of these detections reduces the yearly estimated ozone flux by ~15% at Macquarie Island and ~20% at Melbourne."

In my opinion, Figures 4, 5 and 6 are more supportive-descriptive without adding any-thing new. Therefore, I suggest including them as a supplement. Moreover, Figures 5 and 6 can be merged into one.

As suggested, these figures and accompanying text have been moved into a supplementary document.

Page 11, line 17: "We use the ERA-I 500 hPa data to subjectively classify the events based on their likely meteorological cause." Do the authors classify the events by visual inspection of the 500 hPa maps for every STT event date?

Yes, we looked at each image, because we wanted to determine whether any clearly discernible pattern or dominant weather system connected to the events. At page 10 line 19 we've added: "... likely meteorological cause, by visually examining each date where an event was detected" to the text.

Page 11, lines 20-21: "The stratospheric polar vortex may create ozone folds without other sources of upper tropospheric turbulence". Please include a reference for the above statement.

The sentence now references Baray et al., 2000 and Sprenger et al., 2003.

Page 14, lines 16-20: "The seasonal distributions . . . first half of the year". To my understanding Fig. 7 and Fig. 8 are not quite similar. Moreover, comparing Fig. 8 with Fig. 7 where fire influences are also included is somehow unfair. The fact that ozonesondes are launched monthly at Davis from December to June is also the case for Fig. 7, where high STT frequencies are found for the respective period.

This earlier wording was reflecting that a lower proportion of measurements were made during these months compared with the remainder of the year. In particular, discounting 2011 and 2012 where ozonesondes were launched weekly throughout the year, and 2013 when not flights were made in the early part of the year, the remaining months in summer and autumn have only about 5 flights per month in total. More importantly, we have reconsidered the analysis of STT based on use of the stability criterion, where we now require the square of the Brunt-Väisälä frequency ($N^2$) to be higher at the lapse rate tropopause than the ozone tropopause. This situation allows intruded air to reduce its stability through thermal processes, and discriminates against a small number of cases where intruded air is more stable that the air at the thermal tropopause (which could be the case

when convective air parcels not associated with the STT are near the thermal tropopause). The correspondence between the seasonal distributions between the STT proxy and Fourier analysis are now in closer agreement for Macquarie Island and Melbourne. For Davis, we still have a lack of events for the STT proxy in summer and autumn, and have revised the final sentence of the section that intruduces the STT proxy as follows on page 13, line 11:

"… For our STT proxy, we only detect intrusions where the lowest altitude of the intrusion satisfies the ozone tropopause definition. During summer and autumn, the vertical ozone gradients at Davis are weaker compared with the other seasons, and the detected ozone tropopause tends to lie above the lapse rate tropopause potentially reducing the ability to identify STT events based on the definition of our proxy."

Additionally, we clarify the resolution of the data used. The radiosonde data are binned at 100 m resolution (not 500 m as originally stated), and "$N^2$ is evaluated using 250 m resolution data (to smooth variability in the vertical gradient of potential temperature that is due to small temperature fluctuations likely associated with gravity waves)." is now included in the same paragraph.

Page 16: How is the modeled tropospheric column ozone calculated?
GEOS-Chem provides the ozone density (molecules/cm3), vertical column boxheights, and tropopause level. We are using the sum of the boxheight * density for each box below the one containing the tropopause.
We've added to the text at page 15, line 3: "Figure 9 compares the time series of tropospheric ozone columns ($\Omega_{O3}$) in molecules $cm^{-2}$ simulated by GEOS-Chem (red) to the measured tropospheric ozone columns (black). GEOS-Chem outputs ozone density (molecules $cm^{-3}$ ), and height of each simulated box, as well as which level contains the tropopause, allowing modelled $\Omega_{O3}$ to be calculated as the product of density and height summed up to the box below the tropopause level. In both observations and model..."

How is the tropopause defined in the GEOS-Chem results?
We have added this description to the text at page 7, line 18. : "GEOS-Chem uses the tropopause height provided by GEOS-5 meteorological fields, which are calculated using a lapse-rate tropopause definition using the first minimum above the surface in the function $0.03 \times T (p) - \log(p)$, with p in hPa (Rienecker, 2007)."

Page 17, lines 3-4: "Over Melbourne, ozone in the lower troposphere is well represented, but the model overestimates ozone from around 4 km to the tropopause". This
is also seen for Macquarie and should be added to the discussion.
Over Macquarie Island the lower troposphere seems to be slightly underestimated, which is the same as seen over Davis, while ozone above 4 km does show similar overestimation
The following has been added on page 16, line 5" … The model generally underestimates ozone in the lower troposphere (up to 6 km) over Davis, although this bias is less pronounced during summer. Over Melbourne, ozone in the lower troposphere is well represented, but the model overestimates ozone from around 4 km to the tropopause. Over Macquarie Island we see model overestimation of ozone above 4 km, as well as underestimated ozone in the lower troposphere, suggesting that this region is influenced by processes seen at both of our other sites. Also shown is the mean tropopause height simulated by the model (horizontal dashed red line), which is always higher than the observed average, although this difference is not statistically significant. The effect of local pollution over Melbourne during austral summer (DJF) can be seen from the increased mean mixing ratios and enhanced variance near the surface. The gradient of the O3 profiles is steeper in the measurements than the model, at all sites during all seasons. Recently Hu et al. (2017) examined GEOS-Chem ozone simulations and saw a similar overestimation of upper troposphere ozone in the mid southern latitudes when running with GEOS5 meteorological fields."

Page 19: "Figure 14 shows the mean fraction of total tropospheric column ozone (calculated from ozonesonde profiles) attributed to stratospheric ozone intrusions at each site, averaged over days when an STT event occurred." Please explain in more detail how is this fraction calculated.

In order to clarify how we perform this calculation, we have added the following text on page 20, line 8: " … ozone enhancements, based on a vertical integration of the ozone above baseline levels for each ozonesonde where an event was detected. The top panel of Fig. 12 shows the STT ozone enhancements, based on a vertical integration of the ozone above baseline levels for each ozonesonde where an event was detected. The area considered to be 'enhanced' ozone is outlined with yellow dashes on the left panel of Fig. 4. …

First the tropospheric ozone column is calculated, then the enhanced ozone column amount is used to determine the relative increase."

Page 19: "to the entire Southern Ocean region, defined here as 35_ S-75_ S to en-compass". What is the longitudinal range?

In our extrapolation we used the entire band from 35S to 75S (ie. 180W to 180E).

However, following comments from other reviewers, we have replaced this entire Southern Ocean region with three smaller regions each covering the ozonesonde release sites, please see the new Figure 1.

Page 20: Fig. 14 and Fig.15 can be merged into one.

These images have been merged into one as suggested

Page 22: "If we we assume a fractional ozone impact due to each event STT event of I=35% based on their results". The 30-40% stratospheric contribution found by Terao et al. (2008) is seen only during spring and at 500 hPa. Therefore, assuming a 35% stratospheric contribution to the tropospheric column ozone seems a bit arbitrary.

We agree with the reviewer that this was arbitrary and so we have removed from the revision. We have updated how the calculation of flux is made, and are no longer using this change of I. The updated calculations are on page 19, line 20: "... To determine the ozone column attributable to STT, we determine monthly averaged STT impact (I; fraction of tropospheric ozone sourced from the stratosphere as shown above) and the monthly mean tropospheric ozone column (from the GEOS-Chem multi-year mean, $\Omega_{O3}$ ) over the regions described above. This can be expressed simply as the STT flux per event (flux I in each month: $flux_i = \Omega_{O3} \times I$. Next we determine how many events are occurring per month by assuming only one event can occur at one time, and that no event is measured twice. ...".

**Minor comments:**

We have implemented the following comments as suggested, or else noted here what we have done.
Page 1, line 4: seasonality -> seasonality of STT events
Page 1, line 9: 2.5 km, 3 km -> 2.5 km and 3 km
Page 1, line 14: these -> which
*Page 2, line 2: .Despite lingering -> . Despite the lingering*
Page 2, line 29: found STT -> found that STT
Page 2, line 31: challenging to accurately represent, and better model resolution → challenging to be accurately represented, and finer model resolution
Page 3, line 6: low -> lower
Page 3, lines 14-16: Add references.
Page 3, line 16: characterized -> described
Page 8, line 12: transported -> transported over
Page 9, lines 22-23: (e.g., Sinha et al. (2004); Mari et al. (2008)). -> (e.g., Sinha et al., 2004; Mari et al., 2008). Please check the manuscript for similar instances.

Page 10, line 16: our three sites -> the three sites
Page 10, line 16: detected -> the detected
Page 11, line 23: profile -> vertical profile
Please replace all instances of "Brunt-Viäsälä" in the manuscript with "Brunt-Väisälä".
Page 19, Figure 13: dash -> red dash, please also provide information about the black dashes.
Caption line has been altered to "... GEOS-Chem and ozonesonde pressure levels are marked with red and black dashes respectively"
Page 22, line 9: If we we assume -> If we assume
Page 22, line 10: impact due to each event STT event -> impact due to each STT event
Page 22: empirically-derived threshholds -> empirically-derived thresholds
Page 22: Comparison with ERA-Interim reanalysis data -> Analysis of the ERA-Interim reanalysis data

**Anonymous Referee 2**

**Notes**

The authors present an observation-based method to estimate the total stratospheric ozone flux in the Southern Ocean. I think the approach is interesting and complement some model-based methods, and is also of interest to the readership of ACP. However, the method comes with some major uncertainties and I wonder whether an extrapolation to the whole Southern Ocean from only three measurement sites is reasonable. My major concerns are listed below, and based on them I only recommend the manuscript ready for publication in ACP if a carefully revised manuscript is provided.

**Major Concerns:**

1. **Extrapolation to Southern Ocean**: The authors look at three measurement sites (Davis, Macqaurie, and Melbourne) in the Southern Ocean (SO), and then extrapolate their results to the whole SO. I don't think that this is valid. I think there ia quite a lot of spatial and temporal variability that gets neglected in doing so. To make my point more clearly, I copy a figure (Fig. 16) from Skerlak et al. (2014) here:
It shows the seasonally averaged STT ozone flux for the period 1979-2011. Evidently, there is a lot of spatial and temporal variability. The next figure (Fig. 17) from Skerlak et al. (2014) shows the estimated ozone flux into the PBL, which exhibits a still stronger variability. Hence, I think the authors must be rather hesitating in extrapolating their results. I suggest to restrict the conclusions about the STT flux more to the regions around the three measurement sites. It will still be possible to compare the values, e.g., with the values in Skerlak et al. (2014).
We agree with this comment and as such, in the revision we have removed the Southern Ocean extrapolation from the manuscript. We adopt the reviewer's suggestion of using smaller, more local regions: we examine three regions surrounding each ozonesonde launch site as shown in a new Figure 1. Text has been added at page 19, line 15 to reflect these changes:
"...  Three regions are used to examine possible STT flux over a larger area using modeled tropospheric ozone concentrations. The regions are shown in Fig. 1. The regions are centred at each site, plus or minus ten degrees latitude, and plus or minus 25, 16, and 11 degrees longitude for Davis, Macquarie Island, and Melbourne respectively. These boundaries approximate a rectangle centred at each site with ~2000 km side lengths, covering ∼ 4.4, 4.6, and 4.8 million square km, for Davis, Macquarie Island, and Melbourne respectively."

2. **Transport aspect**: An aspect that is not sufficiently discussed in the manuscript is the transport of the ozone-rich air from its crossing to the measurement site. For instance, in Figure 5 the authors show an STT event and the geopotential height field at 500 hPa. A nice cut-off low pressure system is discernible in the geopotential. But it is not clear whether the STT event really occurred below this cut-off. In fact, it could have happened quite a distance away from it and the be advected to this

place. I would argue that the transport aspect become more important if an STT event is detected at middle or lower-tropospheric levels, i.e., when it is rather 'detached' from the tropopause above. As an example, the following study shows that the crossing of the tropopause takes place in the western North Atlantic but an ozone signal is discernible in the profile over western Europe:

Trickl, T. et al. "How stratospheric are deep stratospheric intrusions? LUAMI 2008." Atmospheric Chemistry and Physics 16.14 (2016): 8791-8815.

I think the authors should more carefully discuss this aspect of STT event. Possibly, the do a short literature review dealing with ozone transport and the long-range character of stratospheric intrusions. It would also be interesting, and relevant to this manuscript, how long signals in stratospheric ozone remain discernible in an atmospheric column after the air parcels have crossed the tropopause.

Transport is hard to quantitatively analyse using sparse ozonesonde measurements. We have updated the flux calculation with an assumed STT event lifetime of ~ 2 days, which makes the transport question even more pertinent as we see in Trickl et al. that long range transport is possible in certain conditions within 48 hours. We have now included more literature in the introduction, dealing with transport at page 3 line 11:

"Stratospheric ozone intrusions undergo transport and mixing, with up to half of the ozone diffusing within 12 hours following descension from the upper troposphere (Trickl et al., 2014).

  The long range transport of enhanced ozone can be facilitated by shear upper tropospheric winds, with remarkably little convective mixing, as shown by Trickl et al., 2014, who measure STT airmasses two days and thousands of kilometres from their source. Cooper et al., 2004, also shows how STT advection can transport stratospheric air over long distances, with a modelled STT event spreading from the northern Pacific to the East coast of the USA over a few days."

Additionally we discuss the affects of uncertainty to do with advection upon our flux estimation: "... Transport uncertainty is very difficult to estimate with the disparate point measurements; it's possible that detected events are (at least partially) advected out of the analysis regions, which would mean we overestimate the influx into the region, and it is also possible that we are influenced by STT events outside the regions of analysis."

3. **Uncertainty**: The method comes with quite a few uncertainties! I list some of them:
- P7,L30: "STT events at altitudes below 4 km are removed to avoid surface pollution, and events within 0.5 km of the tropopause are removed to avoid false positives induced by the sharp transition to stratospheric air." → I see the problem with the near-surface STT events. But still, even at this low altitude it could be due to a stratospheric intrusion.

This is one possible false negative which could occur, I have added a note at page 25, line 12: "However, STT events which reach below 4~km are possible and we may have some false negative detections due to the altitude restricted detections."

Further, I expect quite some ozone flux to be across the tropopause without a very clear peak-like structure in the profile. This could, e.g., be the case if the ozone flux is more related to a continuous 'diffusion' of ozone across the tropopause in contrast to an ozone flux going along with a coherent cross-tropopause air streams in distinct weather systems.

The reviewer raises a very important point here, in that some of what we are defining as our 'background' tropospheric ozone may in fact be a diffuse ozone of recent stratospheric origin, with vertical scales exceeding our bandpass filtering limits. While our work focuses on the strengths of ozonesonde data (namely, their very high vertical resolution), we recognise that we are ill-placed to capture these type of STT ozone flux events noted by the reviewer. This is, in part, due to the low temporal resolution (weekly ozone flights). To make it clear that we are likely missing some of these type of STT events, we have added text at page DOLAST: 8, line 10 which states: *"We note that this ozone detection methodology detailed above does not allow us to resolve STT events where the ozone flux is spread diffusely across the troposphere without a peak-like structure in the ozonesonde profile. In other words, STT events which might have occurred some distance and time away from the location of the ozonesonde profiles may not be readily detected using the high*

*vertical resolution, but infrequent, ozonesonde launches.”*

- P7,L9-12: “This estimate is conservative because it does not take into account any ozone enhancements outside of the detected peak that may have been caused by the STT, and also ignores any enhanced ozone background amounts from synoptic-scale stratospheric mixing into the troposphere.” → The ozone background is also enhanced in mixing across the troposphere, or the background at any of the stations is enhanced by STT events taking place outside its 'range'.

We have noted  this possibility in our revision at page 19, line 9: “... increased the local background mixing ratio, and any influence from STT events nearby which may also increase the local background ozone.”. Also we mention this aspect in our response to the comment above.

- In section 5 (P19,L9) the overall ozone flux is determined as the product of the monthly likelihoods of STT (f), the monthly mean fraction of an ozone column attributed to stratospheric ozone (I) and the mean tropospheric ozone column (Omega). All these factors come with a lot of uncertainty! Be it due to the method applied, or the spatial and temporal variability.

Following advice from another reviewer, we have removed this extrapolation from the revisied manuscript. Instead, we focus on three smaller regions which are shown in a new Figure 1, located around the sites of the ozonesonde launches. The monthly probabilities of detection have been renamed to P, and a new term representing assumed event longevity has been added. The uncertainties in the smaller extrapolations have now been somewhat quantified by using the multi-year standard deviations of each term in the multiplication, shown on page 25, line 3: “Flux is calculated as $I \times P \times M \times \Omega_{O3}$ , with each term calculated as described in Sect. 5.1. The uncertainty is determined using the standard deviation of the product, with variance calculated using the variance of a product formula, assuming that each of our terms is independent:
...
Uncertainty in assumed event lifetime is set at 50%, as we believe it is reasonable to expect events to last 1-3 days. P is the probability of any ozonesonde detecting an event, and is assumed to be constant (for any month). The overall uncertainty as a percentage is shown in parentheses in Table 3, these values are on the order of 100%, largely due to relative uncertainty in the I factor which ranges from 50-120% for each month.”

We also mention the shortfalls of our uncertainty calculations in section 6.2 at page 25, line 27: “...Other possibly important uncertainties in our calculation of STT flux which we don't cover are listed here. Filtering events which occur within 500 m of the tropopause may also lead to more false negatives. This could also cause lower impact estimates due to only measuring ozone enhancements which have descended and potentially slightly dissipated. On the other hand we have no measure of how often or likely the detached ozone intrusion reascends into the stratosphere, which would lead to a reduced stratospheric impact. The estimated tropospheric ozone columns modelled by GEOS-Chem may be biased, for instance Hu et al. (2017) suggest that in general GEOS-Chem (with GEOS-5 met. fields) underestimates STT, with ~360 Tg $a^{-1}$ simulated globally, compared to ~550 Tg $a^{-1}$ observationally constrained. Transport uncertainty is very difficult to estimate with the disparate point measurements; it's possible that detected events are (at least partially) advected out of the analysis regions, which would mean we overestimate the influx into the region, and it is also possible that we are influenced by STT events outside the regions of analysis. Uncertainty in event longevity is set to 50%, however this implies a very simplistic model of event lifetimes, a great deal of work could be done to properly model the regional event lifetimes however this is beyond the scope of our work.”

- P9,L16: “While ozone production occurs in some biomass burning plumes, this is not always the case; therefore ozone perturbations detected during transported smoke events may or may not be caused by the plume. For this reason all detected STT events found near smoke plumes are flagged.” → These events are not included in the calculation of the ozone

flux, but still they could be of relevance!

This is now noted at page 25, line 21: "Our STT event impact estimates have some sensitivity to our biomass burning filter: including smoke-influenced days increases the mean per area flux by 15-20%. Although events which are detected near fire smoke plumes are removed, some portion of these could be actual STTs. The change in our P parameter when we include potentially smoke influenced events leads to a yearly estimated STT of $11 \times 10^{17}$ molecules cm$^{-2}$ yr$^{-1}$ over Melbourne, which suggests that up to $2.1 \times 10^{17}$ molecules cm$^{-2}$ yr$^{-1}$ ozone enhancement could be caused by smoke plume transported precursors. This is a potential area for improvement, as a better method of determining smoke influenced columns would improve confidence in our estimate.."
which also shows the calculated affect of removing potential smoke events.

- P9,L7-9: " We use the 99th percentile because at this point the filter locates clear events with no obvious false positives. Event detection is highly sensitive to this choice; for example, using the 98.5th percentile instead increased detected events by 10 (22%) at Davis, 19 (40%) at Macquarie Island, and 24 (33%) at Melbourne." → Does this mean that with a 98.5 th percentile, some of the events are clear false positives? Wo do you decide that this is the case? I am not sure whether this is obvious. In short, an additional uncertainty of the method.

The sensitivity to this threshold is challenging, we were seeing some false positives when we relaxed to the 98.5$^{th}$ percentile however with the other changes made to our STT detection algorithm as part of addressing the reviewers' comments (such as using the ozone tropopause), we find that the 95$^{th}$ percentile is the optimal balance between maximising data acceptance while minimising false positives (from visual inspection, less than 5%) Given all these uncertainties, the estimate of the total STT flux based on the ozone profiles must be
rather conservative and going along with a big overall uncertainty! This is already discussed by the authors, i.e., they are fully aware of it. What I would, however, suggest is a separate section (or extended paragraph) where all uncertainties are presented and, if possible, quantified.

We have added a Sensitivities and limitations section (Sect. 6) with the uncertainty calculations for both detections and flux estimation. Overall uncertainty in flux is calculated to be around 100%, and we discuss both how we determine this number and the potential uncertainties which we are not able to quantify.

**Minor comments:**

We have implemented the following comments as suggested, or else noted here what we have done.
P2,L13-15: "While models show decreasing tropospheric ozone due to stratospheric ozone depletion propagated to the upper troposphere through vertical mixing (Stevenson et al., 2013), recent work based on the Southern Hemisphere ADditional OZonesonde (SHADOZ) network suggests increasing upper tropospheric ozone near southern Africa, most likely due to stratospheric mixing (Liu et al., 2015; Thompson et al., 2014)" → Simplify sentence structure! Rephrase.
This now reads: "Models show stratospheric ozone depletion has propagated to the upper troposphere (Stevenson et al., 2013). However, work based on the Southern Hemisphere Additional OZonesonde (SHADOZ) network suggests stratospheric mixing may be increasing upper tropospheric ozone near southern Africa (Liu et al., 2015; Thompson et al., 2014)."
- P2,L24: " excedes" → "exceeds"
- P2,L29: "STT is responsible" → "STT to be responsible"
- P3, L4: "mixing across the tropopause mainly caused by the jet streams" → a little strange formulation. Mixing is not caused by the jet streams; maybe you can write that it is associated by the jet streams.
This now reads: "... mixing across the tropopause mainly associated with the jet streams over the ocean."
- P3,L10-11: "A big influence on the high surface ozone concentrations over the eastern Mediterranean is stratospheric mixing and anticyclonic subsidence (Zanis et al., 2014)" → "A big

influence on the high surface ozone concentrations over the eastern Mediterranean can be attributed stratospheric mixing and anticyclonic subsidence (Zanis et al., 2014)"

- P3,L11-12: The authors might want to consider the following studies dealing with STT and ozone fluxes over the eastern Mediterranean:

Tyrlis, E., B. Škerlak, M. Sprenger, H. Wernli, G. Zittis, and J. Lelieveld (2014), On the linkage between the Asian summer monsoon and tropopause fold activity over the eastern Mediterranean and the Middle East, J. Geophys. Res. Atmos., 119, 3202–3221, doi:10.1002/2013JD021113.

Akritidis, D. et al. "On the role of tropopause folds in summertime tropospheric ozone over the eastern Mediterranean and the Middle East." Atmospheric Chemistry and Physics 16.21 (2016): 14025-14039.

- P3,L14-15: "The strength (ozone enhancement above background levels), horizontal scale, vertical depth, and longevity of these intruding ozone tongues vary with weather, topography, and season." → This is a rather general statement. What do you mean with weather?

This line has been updated to : "... vary with wind direction and strength, topography, and season."

- P3,L30-33: How relevant is it for the reader to know how the ozone mixing ratio is quantified? If not relevant, I would remove this sentence. It sounds rather technical!

The technical portion has been removed and the sentence now reads: "... Ozone mixing ratio is quantified with an electrochemical concentration cell, using standardised procedures when constructing, transporting, and releasing the ozonesondes (http://www.ndsc.ncep.noaa.gov/organize/protocols/appendix5/)."

- *P4,L8-9: "Characterisation of STT events requires a clear definition of the tropopause. The two most common tropopause height definitions are the standard lapse rate tropopause (WMO, 1957) and the ozone tropopause (Bethan et al., 1996)." → I would mention already at this place the dynamical tropopause which is defined by means of a potential vorticity iso-surface. I would guess it to be rather similar to the ozone tropopause, but to differ from the WMO one.*

At the end of this paragraph, at page 5, line 18, the following sentence has been added: " Another commonly used tropopause definition (the dynamical tropopause) is determined from the ±2 PVU isosurface, which allows a 3D view of folds and other tropopause features in a sufficiently resolved model (Škerlak et al., 2014)."

- P4,L17: "The ozone tropopause can be less robust during stratosphere-troposphere exchange;" → What does 'robust' mean? What defines whether a tropopause is robust or not? The ozone tropopause certainly allows for much more details (and a much more complicated structure) than the WMO tropopause. But I would not say that it is less robust because of this!

Here we meant robust to mean 'less likely to misdiagnose the tropopause altitude'. As this is unclear, we have changed the text at page 5, line 15: "The ozone tropopause may misdiagnose the real tropopause altitude during stratosphere-troposphere exchange; however, it is useful at polar latitudes in winter, where the lapse-rate definition may result in artificially high tropopause values (Bethan et al., 1996; Tomikawa et al., 2009; Alexander et al., 2013)"

- P4,L19-21: "In this work, the lower of these two tropopause altitudes is used. This choice avoids occasional unrealistically high tropopause heights due to perturbed ozone or temperature measurements in the ozonesonde data." → I feel a little uncomfortable by this definition! The two definitions of the tropopause are rather different, and by simply taking the lower one seems 'dangerous'. The authors should motivate this approach more clearly. At least, I would like to know how often the ozone tropopause 'wins' and how often the WMO one. I would expect the ozone tropopause most often to be at lower heights than the WMO one! Correct?

We now use the ozone tropopause exclusively. We saw several misdiagnosed tropopause heights when using the lapse-rate definition due to low lying temperature inversions.
Actually we tend to see the ozone tropopause at higher altitudes (higher median can be seen in Fig. 3).

- P7,L22-23: "The interpolated profiles are then bandpass-filtered using a Fourier transform to retain perturbations with vertical scales between 0.5 km and 5 km (removing low and high

frequency perturbations)" → I see the 0.5-km threshold. What exactly is the aim of the low-pass filtering threshold (5 km). A more clear description would be helpful.

We select a low-pass limit to remove any background ozone which might, for instance, be slowly increasing with height and could otherwise result in a false STT flag. As noted by other reviewers, our ozonesonde method therefore cannot determine STT which might have occurred some time and space away from the ozonesonde flights because the resultant ozone becomes smeared out through (part of) the troposphere and does not in these cases produce a clear perturbation signal.

- P7,33-34: "The STT event is confirmed if the perturbation profile drops below zero between the ozone peak and the tropopause" → Why does have to drop below zero?

The drop represents a return to non-enhanced ozone concentrations, which suggests separation between the ozone event and the tropopause. We've updated the text to read: "The STT event is confirmed if the perturbation profile drops below zero between the ozone peak and the tropopause, as this represents a return to non-enhanced ozone concentrations."

- P8,Figure 3: Just for curiosity: In the ozone profile the Ozone mixing ratio (OMR) is rather low right above the identified STT event. The OMR is higher than immediately below the STT event. Is their a simple reason why the OMR is so low right above the STT peak?

It could be due to relatively clean free tropospheric air being advected over the event, or else the ozone rich air has been advected into the path of the ozonesonde while the free troposphere was particularly clean.

It's also worth noting that the x axis did not begin at zero molecules per cubic centimetre, and has since been updated to ppbv.

- P9,L16: "all detected STT events found near smoke plumes are flagged." → What does 'near' mean?

'Near' is defined subjectively as within 150km, which has been added to the text.

- P 10,L15-16: "Data from the European Centre for Medium-range Weather Forecasts (ECMWF) Interim Reanalysis (ERA-I) (Dee et al., 2011) product are used for synoptic-scale examination of weather patterns over our three sites on dates matching detected STT events" → Please rephrase! For instance: "Synoptic-scale weather patterns are examined based on the ERA-Interim dataset (Dee et al., 2011). More specifically, the ERA-I products over the three sites are used on dates matching detected STT events.

This sentence has been restructured as the reviewer has suggested page 10, line 16 : "Synoptic scale weather patterns are examined using data from the European Centre for Medium-range Weather Forecasts (ECMWF) Interim Reanalysis (ERA-I) (Dee et al., 2011). This is done using the ERA-I data products over the three sites on dates matching the detected STT events."

- P11,L17+26: Here, the STT event is subjectively linked to a meteorological feature, a cut-off low-pressure system. The argument is not very 'strong'. I don't think that a lowering of the tropopause itself can explain the flux of stratospheric ozone. It would be interesting to see a vertical cross section the cut-off low, with tropopause height included. Is the cut-off low eroded away from below, or how does the flux across the tropopause in the cut-off low really takes place? Some further thoughts on this might be helpful. The following paper might be a starting point: Stohl, A., et al. "Stratosphere - troposphere exchange: A review, and what we have learned from STACCATO." Journal of Geophysical Research: Atmospheres 108.D12 (2003).

We would like to examine the meteorological event relationships in more detail however we feel that this could best be addressed in a future publication and stress that the relationships noted here are subjective. The case studies in this work and some of the discussion about them has been moved into a supplementary document.

**Anonymous Referee 3**

**Notes**

The paper by Greenslade and coauthors presents an analysis of ozone soundings from three locations between -31 ◦ S and -69 ◦ S over several years. The authors analyse the profiles for ozone enhancements in the troposphere, which they link to stratosphere-to-troposhpere exchange associated with cut-off lows and cyclones. Based on these enhancements they estimate the fraction of excess ozone from the stratosphere as percent of the tropospheric ozone column. They calculate a flux of ozone for the southern hemisphere by using tropospheric columns from GEOS-Chem times this fraction times the frequency of occurrence of STT events. This flux estimate is more than an order of magnitude lower than estimates from literature using various techniques. The authors conclude that these differences are due to conservative estimates of several thresholds and assumption regarding their method.

Observationally based estimates of ozone transport especially in the southern hemisphere are sparse and therefore valuable. However, the authors provide a flux estimate which is far from other studies, probably due to the sparse spatial and temporal coverage. If this however is the case, the method is simply not applicable in this case and the deduced flux does not mean anything. If the method is valid for the given data set I miss a careful analysis of the reasons for the discrepancy. Therefore I don't see the paper as an ACP paper in the current form.

**Major comments:**

1) Overall the manuscript leaves me a bit puzzled, since I'm not sure what to take out of this work. The authors state that their value of the ozone enhancement fraction might be largely underestimated by a factor of ten. If this is the case it is difficult to get the benefit of the study. Though the approach is reasonable, maybe the statistics and spatial coverage is to small to cover the full variability and frequency of occurrence of STT events for a quantitative flux calculation for the southern hemisphere. If the difference between observations and models is really between factors 30-200 depending on the reference (p.21, l.8-12), this needs more clarification than simply replacing observation with model results to find agreement with other studies. I found this approach at least very questionable.

We aim to present the ozonesonde dataset along with a method of detecting STT ozone intrusions which takes advantage of the very high vertical resolution nature of the ozonesondes. STT flux estimation was included as a novel use for the ozonesonde dataset, although we agree with reviewers' comments that the extrapolation over the southern ocean was too simplified. As such, we have restricted the calculations to regions adjacent to each ozonesonde launch site, as illustrated in the new Figure 1. In the revised manuscript, we provide more analysis of uncertainties and a better comparison between our results and the literature.

2) Further: I missed a quantification of the uncertainties. This is partly done in section 2.5, but it is e.g. not clear why a threshold of 99% is the best choice nor which factor specifically leads to the very low flux estimate. I suggest the authors use a bunch of northern hemispheric sondes with higher spatial and temporal density to gauge their approach before applying it to the southern hemisphere.

Section 6 has been added which discusses in detail both the uncertainties we calculate and those we are unable to quantify. The 99[th] percentile is replaced by the 95[th], which, following improvements to our algorithm upon reading the reviewers comments, has fewer obvious false positives (less than

5%) detections, which is compensated for by the increased number of detections. We find the 95th perecentile level to be the optimal balance between minimising false positives and accepting more ozone events. This is added to the text at page 24, line 2: "The cut-off threshold (defined separately for each site) is determined from the 95th percentile of the ozone perturbation profiles between 2 km above the earth's surface and 1 km below the tropopause. We use the 95th percentile because at this point the filter locates clear events with fewer than 5% obvious false positive detections. Event detection is sensitive to this choice; for example, using the 96th, and 97th percentile instead decreased detected events by 2, 9 (2,10%) at Davis, 13, 31 (11, 28%) at Macquarie Island, and 8, 24 (6, 18%) at Melbourne."

This updated and improved analysis procedure is more robust than that described in our submitted manuscript. We think that the reviewer's suggestion to use a dense Northern Hemisphere network of ozonesondes is valuable and we will consider pursuing this in a separate study.

**Minor comments:**

We have implemented the following comments as suggested, or else noted here what we have done.
p.1,l.5: Please add the period of observations
p.4, l. 9: At least mention the dynamical tropopause, it is more common than ozone...
We have added the following at page 5, line 18 "... Another commonly used tropopause definition (the dynamical tropopause) is determined from the ±2 PVU isosurface, which allows a 3D view of folds and other tropopause features in a sufficiently resolved model (Škerlak et al., 2014)."
p.4, l.11: Correct definition of the thermal tropopause "... provided the lapse rate averaged between this altitude …"
C2p.4, l.20 (also Fig.1): The tropopause definitions are mixed here. Why do the authors not include the dynamical definition? The effect of the pure lapse rate criterion is misleading under specific synoptic conditions as correctly stated. This might explain the very low cases in Fig.1.
This is indeed the cause of the low tropopause detections: the lapse-rate definition has been fixed in the latest version to exclude detections below 4km, which were all due temperature inversions near the boundary layer. The revised Figure 2 now shows the multi-year ozone tropopause altitude climatology.
The following text was added at page 5, line 17: "... We require lapse-rate tropopauses to be at a minimum of 4 km altitude."
Regarding the dynamical tropopause, using solely the sonde data we lacked sufficient information to determine the PV, and we wanted to keep the analysis of sonde records unmodified by other datasets (such as modelled PV).
p.6, l.15: How many model levels are between the sea level and 14 km? How many model levels are between 8 and 14 km and how are sonde and profile data compared? Pointwise or vertically averaged to fit the model levels?
Model and sonde datasets are only compared using the vertically summed tropospheric ozone columns [molecules / cm$^2$].
Vertical model resolution is roughly 60 m near the surface, and around 500 m near 10 km altitude, which has been added to the text at page 7, line 14: "The vertical resolution is finer near the surface at ~60 m between levels, spreading out to ~500 m near 10 km altitude."
p.6, l.15+: The sonde profiles are compared against model data of 2 x 2.5 degrees grid sizes (and the vertical model resolution). How well does the model resolve the soundings?
Generally not too well, but we do see an agreement between the datasets in terms of season and amplitude. We show some examples of the comparison between model and ozonesondes in Figure 14: clearly the model struggles to reproduce the short vertical features in ozone recorded by the ozonesondes.
How do the authors estimate the fraction of ozone transport which is missed due to unresolved

structures?

We assume that if the structure is unresolved, we cannot be certain that it is an STT event. At this time we have not examined the likelihood and frequency of false negatives.

Why do the authors don't interpolate to the time window of the sounding (or at least use the according model time step)?

We do use the closest matching time steps for comparison, although we did not make this point clearly in the manuscript. The daily model time step over Davis is 0100, 0700, 1300, 1900, of which 0700 is generally closest to sonde release times. This is due to the model using a globally instantaneous (rather than local) snapshot.

p.11, l.17: Even if you did a subjective method, could you explain a bit more in detail in the manuscript, how you distinguished different potential situations? What are upper tropospheric "low pressure fronts"? Tropospheric intrusions (3D!) in the stratosphere or stratospheric cut-offs (fully detached)?

The case studies showing two of these events has been moved into a supplementary document following a suggestion from Referee one, and the following has been added to the text: page 11, line 28: "... To detect cut-off low pressure systems we look for cyclonic winds and a detached area of low pressure within ~500 km of a site on days of event detection. For low pressure fronts we look for low pressure troughs within ~500 km."

p.11,l.20: What are "ozone folds" without other sources of upper tropospheric turbulence and how are these related to the polar vortex?

We meant to say tropopause folds, as this tropopause folding and increased stratospheric mixing is seen near many jet streams. The sentence has been changed to "… The stratospheric polar vortex may create tropopause folds without other sources of upper tropospheric turbulence such as low pressure fronts or cyclones (e.g. Baray et al., 2000; Sprenger et al., 2003; Tyrlis et al., 2014) ."

p.11, l.25: Explain: "...ozone enhancements derived from dry stratospheric air..." didn't you use the methods and criteria from sec.2?

Derived was poor word choice, we meant simply that the ozone enhancement was likely due to stratospheric influx. The sentence has been changed to "... suggesting the ozone enhancements **are due to** dry stratospheric air."

p.12, Fig.6. and related discussion (shortly before section 3): Please show a cross section of PV since most likely the ozone peak is related to a tropopause fold.

We have removed this figure and discussion in the revision.

p.12, last line: What is meant with increased winter activity? More tropopause folds, stronger tropospheric winds, cyclone activity, etc...? Please be more precise. How do you expect the vortex to affect the tropopause?

We were commenting on the relative increase in STT event detections over Davis compared to the other sites, the text has been altered to read: "… At Davis, there are more STT detections during winter relative to our other sites, which may be due to the polar vortex and its associated lowered tropopause and increased turbulence."

p.14, l.6-20: Why do you use N2 as indicator? The relation you found is interesting, but not necessarily valid since stability is not conserved. Why should it be 'retained' when crossing the thermal tropopause?

We use Brunt-Väisälä frequency as an indicator of transport as it has characteristically different behaviour in the stratosphere and troposphere and can be directly evaluated from the radiosonde measurements. We agree that static stability is not conserved when crossing the tropopause. As discussed in the manuscript, for this particular analysis we require the transport time to be short (hours to 1-2 days) compared with the radiative time. This is generally the case in the extratropical tropopause (Gettleman et al., 2011 - http://onlinelibrary.wiley.com/doi/10.1029/2011RG000355/pdf), although we are necessarily biased identifying STT that satisfies this case.

In general the thermal tropopause is ill defined under these conditions. Why not simply taking PV for this excercise or humidity as a measurement based quantity?

As we were seeing problems with the lapse-rate tropopause, and in response to several comments regarding the tropopause definitions, we now use the ozone defined tropopause exclusively for event detection.

Regarding PV: PV is necessarily only available at lower vertical resolution from reanalyses. The radiosonde humidity is biased low near the tropopause, and the performance of the humidity sensor degrades with lower temperatures, giving poorer performance in winter compared with summer.

Fig.11 and related discussion: Couldn't you provide scatter plots (or Taylor diagram) of the column ozone between sondes and model?

We prefer to display the model and ozonesonde as time-series (now Figure 9 in the revision) to allow us to comment on the seasonality.

Fig.3 caption: Units: concentration or mixing ratio?

This image has been updated to use ozone ppbv for both panels, with the caption updated.

---

## Author Response (AR2)

We have made all the corrections listed below, with a few additional responses provided in blue.

*Report 1:*

In authors' responses to my first review many of my comments have been addressed. I believe that the revised manuscript is improved both in terms of scientific approach and structure. Nevertheless, I still see place for comments. Moreover, I don't see any improvement with the language usage in the revised version. I would recommend publication of the manuscript after the authors improve the readability of the manuscript and address the following minor comments.

- Page 1, line 8: above all three sites -> for all three sites
- Page 1, line 9: from STT events -> owing to STT events
- Page 1, line 9: and Macquarie Island -> and Macquarie Island, respectively
- Page 2, line 9: show stratospheric ozone -> show that stratospheric ozone
- Page 2, line 21: Another important region of STT -> Another hotspot of STT
- Page 2, lines 21-22: with 10 ppbv from STT contribution -> with a stratospheric contribution of 10 ppb
- Page 2, lines 32-33: "Lower stratospheric ... over the ocean". Which ocean? Please rephrase.
- Page 3, lines 5-7: Replace sentence with: "The summertime pool of high tropospheric ozone over the eastern Mediterranean (EM) is mainly attributed to the downward ozone transport, as a result of the enhanced subsidence (Zanis et al., 2014) and the tropopause fold activity (Akritidis et al., 2016) over the region."
    Your sentence has been implemented, note that latexdiff cuts off the 'is' even though it is there in the final pdf.
- Page 3, line 7: frequently shows -> exhibits
- Page 3, line 7: stratospheric subsidence -> subsidence
- Page 3, line 13: "shear upper tropospheric winds" -> "upper tropospheric wind shear" ?
    This was meant to say just "upper tropospheric winds", fixed in text. The wind shear was a potential driver of tropospheric mixing rather than transport (Trickl et al. 2014).
- The "southern hemisphere" phrase is first seen at page 2, line 9. You should add the SH abbreviation there instead of the current position in the manuscript.
- Page 3: "Section 2 describes", "Section 3 describes", "Section 4 analyses": I suggest rephrasing these with "In Section 2 we describe.." or something similar.
- Figure 1 caption: I suggest removing "which will be".
- Page 4, line 3: to 35 -> and up to 35
- Page 4, line 4: make -> perform
- Page 5, line 16: high tropopause values -> high tropopause height values
- Page 5, line 17: lapse rate tropopauses -> lapse rate defined tropopause
- PVU is the unit and potential vorticity (PV) the variable. Please replace at page 5 lines 18-20 "Another commonly used...Tyrlis et al., 2014)." with the following: "Another commonly used tropopause definition is determined with the use of PV (dynamical tropopause). In the extra-tropics the isosurface where PV=2 PVU (1 PVU= 10-6 m2 s-1 K kg-1) is often used to define the tropopause, allowing the 3D representation of tropopause folds and other tropopause features in a sufficiently resolved model (Škerlak et al., 2014; Tyrlis et al., 2014)".
- Page 5, line 20: The PVU-> PV
- Page 11, line 18: ozone fold -> tropopause fold
- Page 11, lines 19-20: "Their work seems.. ozonesonde measurements." I don't find any remarkable agreement between your findings and the findings by Skerlak et al. (2015) over Melbourne. In Skerlak et al. (2015) the fold frequency over Melbourne during DJF is 1-2% and

during JJA is 0.5-1%, which in no case agrees with your findings (much higher STT frequency during DJF compared to JJA).

You are right, our comparison to this image was too subjective anyway (since we are just looking closely at images rather than analysing their actual data). The text has been updated from: "... which agrees with our ozonesonde measurements." to **"... however not to the same extent that our summer peak suggests."**

- Page 14, Figure 7 caption: STT event altitude -> STT events altitude
- Page 15, Figure 8 caption: STT event depth -> STT events depth
- Page 15, Section 4 title: I suggest using "Simulated ozone columns" instead of "Simulation of ozone columns".
- Page 16, line 14: and saw -> and found
- Page 16, lines 14-15: when running with -> when using the
- Page 19, lines 22-23: "(fluxi.. ". Close parenthesis.
- Page 20, line 31: use a model with stratospheric ozone tracer -> used a model carrying a tracer for stratospheric ozone
- Page 21, Figure 12: Multiply with 100 the y axis values on the bottom figure in order to be the % fraction.
- Page 27, means they are capable -> suggests they are capable
- Page 27, line 3: tropsopheric -> tropospheric
- Supplement, Figure S1 caption: Melbourne ozone -> ozone over Melbourne

**Report 2:**

The authors performed a major revision of their manuscript, which lead to a significant improvement of the paper. Particularly they changed the method of calculating the ozone fluxes by also accounting for the duration of the events. This led to different values for the ozone flux. Therefore they could remove the rescaling of the event intensity by using the model results of Terao et al., (2008) to get a better agreement with other studies. The new estimates for ozone fluxes are in the range of other estimates (Skerlak et al., 2014) and are compared to this study. They also added a detailed error discussion and a comparison with Skerlak et al., (2014), which shows, that the updated approach leads to reasonable results.
I therefore recommend the paper in the present form with a few technical correction.

Fig.12: Please check the y_axis of the percentage plot (bottom)

p.10, l.27: If we assume…. check the sentence - it seems to be incomplete

[revised manuscript text omitted]